# Benchmarking Bias Mitigation Algorithms in Representation Learning through Fairness Metrics

**Charan Reddy**
Mila, University of Montreal
`reddycha@mila.quebec`

**Deepak Sharma**
Mila, Mcgill University

**Soroush Mehri**
Microsoft Research

**Adriana Romero-Soriano**
SCS McGill University

**Samira Shabanian** *
Microsoft Research

**Sina Honari** *
CVLab, EPFL

## Abstract

With the recent expanding attention of machine learning researchers and practitioners to fairness, there is a void of a common framework to analyze and compare the capabilities of proposed models in deep representation learning. In this paper, we evaluate different fairness methods trained with deep neural networks on a common synthetic dataset and a real-world dataset to obtain better insights on how these methods work. In particular, we train about 3000 different models in various setups, including imbalanced and correlated data configurations, to verify the limits of the current models and better understand in which setups they are subject to failure. Our results show that the bias of models increase as datasets become more imbalanced or datasets attributes become more correlated, the level of dominance of correlated sensitive dataset features impact bias, and the sensitive information remains in the latent representation even when bias-mitigation algorithms are applied. Overall, we present a dataset, propose various challenging evaluation setups, and rigorously evaluate recent promising bias-mitigation algorithms in a common framework and publicly release this benchmark[†], hoping the research community would take it as a common entry point for fair deep learning.

## 1 Introduction

The success of deep learning models and their quick adoption in many application domains has brought up important questions on the fairness of these models when deployed in the real world. Recent studies have highlighted the biases encoded by representation learning algorithms and have questioned the reliability of such approaches to make decisions. In particular, [1] outline the biases exhibited by learning algorithms whose goal is to match the data distribution in an adversarial setting. Similar findings have been revealed in the context of visual question answering [2], image search tasks [3], language models [4] and gender classification [5]. As a result, there is increasing interest in understanding the sources of bias in learning algorithms and developing bias-mitigation strategies [6, 7, 8, 9].

The goal of bias-mitigation algorithms is to mitigate the influence of sensitive data features on the made decisions of eligibility. Sensitive features are private and protected features of a dataset such as

---

*equal advising

[†]Our code is available at `https://github.com/charan223/FairDeepLearning`

35th Conference on Neural Information Processing Systems (NeurIPS 2021) Track on Datasets and Benchmarks.

gender or race, which should not affect output decisions of eligibility. The eligibility is the criteria that makes an individual qualified or unqualified for a given task, such as giving loans or hiring. Bias mitigation models aim at making eligibility decisions on dataset samples without having bias towards the sensitive attributes of the input data. The difficulty of bias-mitigation tasks is often determined by the dataset distribution, which in turn, is a function of the potential label and feature imbalance, the correlation of potentially sensitive features with other features in the data, and the perhaps inevitable distribution shift from training to the development phase, to name a few. We argue that without evaluation of bias-mitigation models in various challenging setups, their merits remain unclear.

In addition to the challenges associated with each dataset and task, the current state of the bias-mitigation literature hinders method comparisons due to inconsistencies in the experimentation and dataset setups. Given the importance and possibly tangible impact of the proposed algorithms when in production, we advocate for a rigorous and unified evaluation protocol to assess the capabilities of bias-mitigation models. We argue the need for a systematic analysis that would benchmark different bias-mitigation approaches under the perspective of different fairness metrics to ensure replication of concluded results. This would help elucidate the most promising research contributions and possible future avenues to explore.

Therefore, this paper aims to provide a unified framework to benchmark bias-mitigation approaches and perform an in-depth analysis of existing approaches leveraging the proposed framework. We start with introducing a synthetic dataset that facilitates creating challenging scenarios by controlling data imbalance or correlation among eligibility and sensitive or non-sensitive attributes. Contrary to real datasets, where the different components of data generation cannot be controlled independently, this synthetic dataset enables the soft modification of dataset characteristics, by *e.g.* changing one component and keeping all others unchanged, which in turn allows the study of different sources and levels of bias in the data. We also consider a real and commonly used dataset, the Adult dataset [10], to investigate the impact of our settings on real data. In this case, we adapt the dataset characteristics by altering the binarization process of its sensitive attributes, obtaining variations of the data reminiscent of those explored in the synthetic dataset we introduced.

We then provide an in-depth analysis of baselines and recent bias-mitigation models, leveraging both datasets mentioned above. In particular, we evaluate three promising bias-mitigation models – and seven variants – together with two baseline models. The analysis is performed by considering a unified set of fairness metrics and reporting results by carrying extensive hyper-parameter search in all cases, ensuring that the drawn conclusions can be attributed to modeling or loss choices. In doing so, we train about 3000 models in increasingly difficult scenarios, in which we transition from balanced dataset setups towards challenging imbalanced and correlated setups, where the eligibility criterion is correlated with sensitive or non-sensitive attributes.

Given the importance of bias-mitigation approaches and the severe implications of their misuse, we intentionally try to push these models to their breaking point by creating various *challenging datasets*. We make the following observations through extensive experiments: (1) when the correlation between the eligibility and the sensitive attribute in the training data increases, models tend to exploit it and become more biased in their predictions. These biases are further accentuated as the sensitive features become more predominant; (2) when a group is under-represented, bias can arise due to imbalance or scarceness of the data, both of which affect existing models in different ways. Moreover, as the under-represented group becomes proportionally more imbalanced, the models act more biased; (3) bias-mitigation models do not *completely* remove sensitive information from their latent representations. Instead, they successfully reduce the bias in their results by designing loss functions that balance different subgroups; (4) the robustness to random seeds is model dependent, with some models exhibiting high variance in their results, making the choice of random seed a source of potential bias.

To summarize, we make the following contributions:

- We provide a dataset with controllable sets of features and correlation levels to facilitate research in bias-mitigation in a wide range of scenarios.
- We propose challenging test setups to evaluate bias-mitigation models by considering increasingly imbalanced and correlated scenarios and perform a rigorous analysis of existing methods, showing there remains much room for improvement.
- We release a benchmarking codebase composed of seven state of the art models and two baselines to the community for reproducible evaluation of bias-mitigation algorithms.

It is noteworthy that our analysis is not to undermine the effectiveness of any bias-mitigation method but instead to set the expectations and boundaries for different use cases and to encourage the community to investigate models under more extensive scenarios.

## 2    Related Works

There is increasing literature studying how biased datasets can bias learning algorithms to discriminate [1, 2, 6, 7, 11, 12, 13]. These recent studies have led researchers to propose new evaluation tools and datasets [5, 14, 15, 16, 17, 18], to identify potential error rate gaps among different groups in machine learning.

Bias-mitigation algorithms can be roughly categorized into pre-processing, in-processing, and post-processing approaches. Pre-processing techniques include reweighting of training samples [19], editing features and labels [20], resampling datasets [21]. Post-processing techniques calibrate predictions given sensitive attributes at inference time [14, 22, 23]. In-processing techniques [6, 7, 8, 9, 23, 24, 25, 26, 27, 28, 29, 30, 31, 32, 33] try to remove sensitive information from the learned representation space. Due to the emergence of more in-processing models in the deep learning community, we focus on this group of models, as bias-mitigation algorithms are mainly applied directly to the learning models.

Deep learning approaches have become more popular in the fairness community. For example, Louizos et al. [25] employs a deep variational autoencoder [34] to learn fair latent representations. At the same time, adversarial learning – introduced initially within the framework of Generative Adversarial Networks (GANs) [35] – has also been extensively leveraged in the bias-mitigation literature [6, 7, 8, 9, 26, 27, 28, 29, 30, 31, 32, 33] to make different groups indistinguishable from one another with respect to a sensitive attribute. Due to more widespread usage of adversarial approaches, we analyse this group of models. Adversarial bias-mitigation techniques can be divided into approaches which: (i) seek to mitigate bias through adversarial training directly applied on the class labels, where the class label is an indicator of eligibility [9]; (ii) focus on mitigating bias through enforcing group fairness on the learned latent space where the latent is directly used for classification of eligibility [6, 7, 27, 32, 33]; and (iii) discard sensitive features for downstream tasks after disentangling the learned latent space into sensitive and non-sensitive features [8, 26, 28]. This line of research is motivated by the recent development in disentangled representation learning [25, 29, 30, 31].

To evaluate bias-mitigation models, some contributions have emerged to probe machine learning systems at different levels and reduce discrimination from the perspective of modeling. Friedler et al. [36] evaluate the fairness of pre-processing, in-processing, and post-processing machine learning approaches. Verma and Rubin [37] evaluate how fair an off-the-shelf logistic regression model is, given a set of fairness metric definitions. Contrary to these works, we evaluate *deep learning-based bias-mitigation models*. In particular, due to the popularity and further usage of in-processing adversarial methods, we take some of the promising approaches from this category and assess their merits in challenging setups. [38] provide a review on sources of bias in deep learning models and different approaches applied to them; however, they do not evaluate and contrast models experimentally. To the best of our knowledge, we provide the first benchmark to compare deep learning-based bias-mitigation approaches systematically in varied and increasingly challenging scenarios, in particular, in imbalanced and correlated dataset setups with the goal of bringing further insights into the working of bias-mitigation methods and proposing new frameworks for evaluating them.

## 3    Methodology

In this study, we empirically analyze the performance of two baselines without any bias-mitigation learning criteria: an Mlp (multi-layer perceptron) and a Cnn (convolutional neural network). We compare these baselines with four variants of Laftr [6] (Laftr-DP, Laftr-EqOpp1, Laftr-EqOpp0, and Laftr-EqOdd), which apply adversarial learning to achieve group fairness by leveraging different fairness criteria, two variants of Cfair [7] (Cfair and Cfair-EO) which perform conditional alignment of representations to achieve accuracy-fairness trade-off, and finally, Ffvae [8], which disentangles latent representations into sensitive and non-sensitive features. Check Section C in Appendix for a complete description of these models.

For simplicity, we introduce the following shorthand notation. We denote by $\mathcal{X} \subseteq \mathbb{R}^d$ and $\mathcal{Y} = \{0, 1\}$ the set of inputs and outputs, respectively. Two sets of random variables $X$ and $Y$ take associated

values $x \in \mathcal{X}$ and $y \in \mathcal{Y}$. Variable $y$ determines eligibility, and we denote the sensitive binary variable by $S$ taking values $s \in \{0, 1\}$. Moreover, $p$ indicates the output probability of the classifier as a real number and $\hat{Y}$ is the class predicted random variable based on $p$, which takes values $\hat{y} \in \{0, 1\}$ using a threshold of $0.5$. Furthermore, the term $\mathcal{D}_s^y$ is the conditional distribution of the joint distribution $\mathcal{D}$ over $X \times Y \times S$, given $Y = y$, $S = s$.

## 3.1 Datasets

We use two datasets in our evaluations: a synthetic dataset, which facilitates a controlled data manipulation and generation, and a real dataset, to verify that the obtained results carry over to real data.

**CI-MNIST .** In order to evaluate the bias-mitigation approaches in challenging setups and be capable of controlling different dataset configurations; we design a variant of the MNIST dataset [39], called *Correlated and Imbalanced MNIST* or in short CI-MNIST , where we introduce different types of correlations between attributes, dataset features, and an artificial eligibility criterion. For an input image $x$, the label $y \in \{1, 0\}$ indicates eligibility or ineligibility, respectively, given that $x$ is even or odd. We define the background colors ($bck$) as the protected or sensitive attribute $s \in \{0, 1\}$, where blue denotes the unprivileged group and red denotes the privileged group.

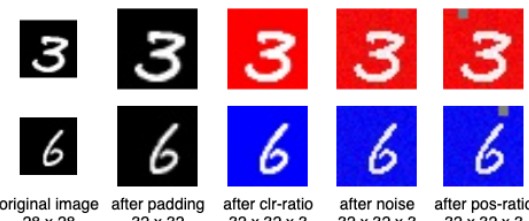

original image    after padding    after clr-ratio    after noise    after pos-ratio
28 x 28    32 x 32    32 x 32 x 3    32 x 32 x 3    32 x 32 x 3

Figure 1: The conversion process used to generate samples. Input image is first padded to become 32x32. The attributes *clr-ratio* is then applied, which decides the background color (blue or red). Noise is then added to the background color. Finally, *pos-ratio* affects the positioning of the box (top row, left half or top row, right half). Check Section D of the Supplementary for more details. Best viewed in color.

The primary motivation for using this dataset is to control different features of the data generation process to simulate increasingly correlated and imbalanced datasets. In doing so, we can analyze models in setups exhibiting different levels of bias and measure their robustness. In particular, we use the following dataset components to create such challenging setups.

clr-ratio: Denoted as $(b_e, b_o)$ pair and used to refer to the percentage of images in the dataset with blue backgrounds in (even, odd) classes. When background is used as a sensitive attribute, the pair $(b_e, b_o)$ indicate the percentage of unprivileged population in (eligible, ineligible) groups, as blue is used for unprivileged group and even and odd indicate respectively eligibility and ineligibility. The rest of the digits in each group have red backgrounds. Using this feature, one can control the correlation between the sensitive attribute (background color) and the eligibility (being even or odd) or create imbalanced datasets between under-represented and over-represented groups.

pos-ratio: Denoted as $(l_e, l_o)$ pair where $l_e$ ($l_o$) refers to the percentage of even (odd) digits in the dataset with a small box in the top-left half of the image. The rest of the digits have the box in the top-right half. Figure 1 depicts image samples with such small boxes. Note that when pos-ratio is $(1, 0)$ the small box is always on the top-left for even digits and always on top-right for odd digits, so its location indicates the eligibility of the sample (being even or odd), yielding the completely correlated setup. On the other hand, when pos-ratio is $(0.5, 0.5)$, for either even and odd digits, in 50% of samples the small box is on top-left and in the other 50% samples it is on top-right of the image, hence its location is not correlated with eligibility, yielding the completely decorrelated setup. By transitioning between these two extreme cases one can control the degree of correlation between eligibility and the position of the box, as a non-sensitive or sensitive attribute, in order to create correlated setups. We do not apply any correlation between small box location and background color.

While training sets can be imbalanced, we use a balanced test set, meaning clr-ratio=(0.5, 0.5) and pos-ratio=(0.5, 0.5) in order to evaluate results on balanced dataset sub-groups and ease comparison between different setups. See Section D in Appendix for complete details of data generation process.

**Adult dataset.** To verify the configurations evaluated on the proposed synthetic dataset carry over to more realistic cases, we use the Adult database [10] given its ubiquitous use in the community [6, 7, 9]. The Adult dataset consists of 30,162 training and 15,060 testing samples of individuals with 112 features such as gender, age, and nationality. Given the features of an adult, the prediction

task is to determine whether the person makes over 50,000 (eligible) or not (ineligible). The binary eligibility value $y$ is defined by salary being over or less than 50,000.

To generate configurations on the Adult dataset with varying complexity, we consider various thresholds to binarize the sensitive attribute age, which is a multi-valued sensitive attribute and is often binarized [40, 41, 24]. Note that we evaluate the aforementioned bias-mitigation approaches on a new Adult test set, which is a subset of the Adult test set but balanced in terms of eligibility and sensitive attributes, similar to our CI-MNIST (see Section D in Appendix for more details). Our goal is to assess different possible configurations of a real-world dataset in similar setups as the ones used in CI-MNIST .

age-ratio: Finding settings in a real dataset that resembles the ones chosen in a synthetic dataset is not trivial since contrary to synthetic datasets the features of a real dataset cannot be controlled for data generation. We binarize age by thresholding it to find correlations between this sensitive attribute and the eligibility in the Adult dataset that are closest to the ones we used in the CI-MNIST dataset. We call this feature age-ratio and show it in pair $(a_e, a_i)$, which refers to the ratio of people who are unprivileged in (eligible, ineligible) classes. The rest of the people in each class are considered as privileged. Refer to the Table 3 in Section D of Supplementary for various thresholds of age used in our experiments.

### 3.2 Evaluation Metrics

We use the same set of fairness metrics, as reported in Laftr [6], Cfair [7], and Ffvae[8] to evaluate these bias-mitigation strategies. In particular, we use Demographic Parity (DP), Equality of Opportunity with respect to label classes $y$ equal to $0$ and $1$ (EqOpp0 and EqOpp1), Equality of Odds (EqOdd), and label accuracy (acc) in our experiments. Please see Table 1 in Appendix for details on the used fairness metrics such as their mathematical formulations and their abbreviations used in the paper. Our framework provides all metrics listed in Table 2; however, due to space limitations, we focus on the metrics mentioned above given their widespread usage.

## 4 Experiments

To evaluate the robustness of the bias-mitigation approaches, we leverage the datasets introduced in section 3.1 and create challenging scenarios where we change the balance of the under-represented group, correlate sensitive attribute and eligibility, and also include scenarios where small features in the image, considered as either sensitive or non-sensitive attributes, are correlated with the eligibility.

In all scenarios, we change the dataset from a balanced setup to different imbalanced setups at training, while at test time, we always evaluate the models in a balanced setup, meaning the dataset is at 50% for eligible and ineligible groups as well as for privileged and unprivileged groups. This way, we report the model's performance on a balanced setup of different sub-groups. On CI-MNIST in the balanced setup, the ratio of unprivileged (blue) and privileged (red) background is at 50% for both eligible and ineligible groups, giving *clr-ratio* of $(0.5, 0.5)$. Moreover, the small-box location and eligibility are uncorrelated with *pos-ratio* being $(0.5, 0.5)$. On Adults, the value of *age-ratios* is $(0.5, 0.5)$ in the balanced setup.

To report results, we initially average metrics over three random seeds, then for each metric, the best value over hyper-parameters is reported on the test set. By doing so, we report the best performance on each metric without choosing how to compromise between accuracy and a fairness metric (which usually have a tradeoff and improving one deteriorates the other). This would allow us to illustrate bias even when the best variation of a model is considered on each metric. Note that this is in favor of the models as the best possible result is reported. Hence each column can correspond to a different hyper-parameter. The details of all datasets, architectures and hyper-parameter setups are provided in Section D of Appendix. Next, we present results in each setting.

**Setting 1. Impact of reducing the representation of the unprivileged group.** We first evaluate the impact of the representation percentage of the unprivileged group in both eligible and ineligible groups. The question we want to ask is whether the models are robust to the small presence of under-represented groups. To obtain such a setup, in CI-MNIST dataset we change *clr-ratio* from the balanced setup of $(0.5, 0.5)$ to imbalanced cases of $(0.1, 0.1)$, $(0.01, 0.01)$, and $(0.001, 0.001)$. In the Adult dataset, we change *age-ratio* from $(0.5, 0.5)$ to the imbalanced cases of $(0.1, 0.1)$ and $(0.01, 0.01)$. Figures 2(a), 2(b) and Figures 3(a), 3(b) compare models trained on balanced and reduced representation data, both on Adult and CI-MNIST datasets. We observe that, through the transition from balanced to imbalanced

setups, for all models both unprivileged group accuracy and fairness metrics drop. This indicates even the bias-mitigation algorithms are susceptible to under-representation of the unprivileged groups. On both CI-MNIST and Adults, we observe the same trend and bias enlarges when the datasets become more imbalanced. Figures 5 and 6 in Section E.1 of Appendix provide results for all dataset configurations. Tables 6 to 13 for Adults and tables 14 to 21 for CI-MNIST report detailed results for each model.

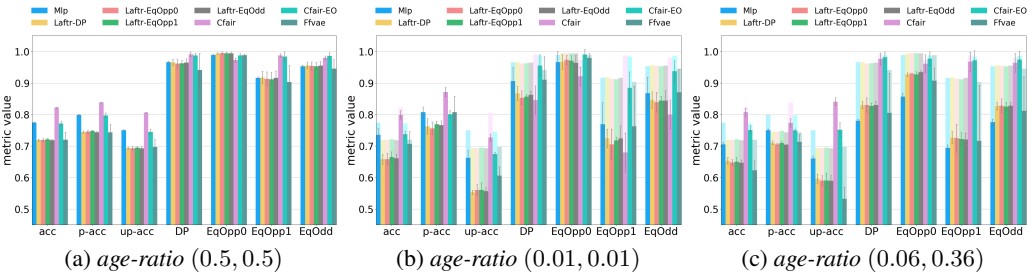

(a) *age-ratio* $(0.5, 0.5)$      (b) *age-ratio* $(0.01, 0.01)$      (c) *age-ratio* $(0.06, 0.36)$

Figure 2: Comparing models on Adults dataset. Sub-figure 2(a) shows the balanced case. Sub-figure 2(b) shows the impact of reducing unprivileged group in Setting 1. Sub-figure 2(c) shows the impact of correlation of sensitive attribute and eligibility in Setting 2. Compare both 2(b) and 2(c) with 2(a) to observe the biased setting's impact. In sub-figures 2(b) and 2(c) the pale colors show the decrease in performance compared to the balanced case in 2(a). Note that in 2(a) the models are not perfect as the performance is not always 1. Higher is better for all metrics. For each bar, the standard deviation is also shown. Best viewed in colors.

**Setting 2. Impact of correlation of sensitive attribute with eligibility.** In this setting, we want to evaluate the resistance of models to the correlation between sensitive attribute and eligibility. The goal is to evaluate whether models perform fairly in the presence of such correlations. To that end, we train models on a training set exhibiting sensitive attribute and eligibility correlations.

To provide such a scenario, in CI-MNIST we change *clr-ratio* from the balanced case of $(0.5, 0.5)$ to correlated cases of $(0.1, 0.9)$ and $(0.01, 0.99)$, and for Adult dataset we change *age-ratio* from $(0.5, 0.5)$ to $(0.06, 0.36)$, where the unprivileged group becomes under-represented in the eligible group and over-represented in the ineligible group, hence correlating eligibility and sensitive features. Note that the setting selected for the Adult dataset is the closest to the selected CI-MNIST setting that we could emulate with this data.

Figures 2(a), 2(c) and Figures 3(a), 3(c), 3(d) compare model performances under different metrics for the Adult and CI-MNIST datasets. For all models, we observe a big drop in both accuracy and fairness metrics on both Adult and CI-MNIST. Note that the bias increases when the correlation level increases from 3(c) to 3(d). The drop is bigger than in the previous evaluation setting, where the unprivileged representation was reduced. This suggests that models are even more susceptible to correlations between eligibility and sensitive attributes. Also note that in both Settings 1 and 2 we observe the same trends in CI-MNIST as in Adults, highlighting the transferability of the observations between datasets. Figures 7, 8 and Tables 22 to 37 in Section E.2 of Appendix present all models and their detailed results.

**Setting 3. Impact of correlation of non-sensitive attribute with eligibility.** In this setting, we want to evaluate to what extent a model can exploit the correlation between non-sensitive and non-predominant attributes and eligibility? To evaluate this setting on CI-MNIST, we keep *clr-ratio* at $(0.5, 0.5)$; however, we change *pos-ratio* from the balanced setting of $(0.5, 0.5)$ to the imbalanced settings of $(0.9, 0.1)$. Note that this would make the position of box highly correlated with eligibility. Figures 3(a) and 3(e) compare different models under this setting. In this case, since *pos-ratio* is a non-predominant non-sensitive feature compared to *clr-ratio*, we do not observe any specific trend in drop of fairness measures among models, but we do see a consistent drop in accuracy metrics, highlighting a slight bias in the models. Note that fairness metrics are measured on the sensitive feature, which in this case is the background color. Figure 9 and Tables 38 to 45 in Section E.3 show detailed results for all models.

**Setting 4. Impact of correlation of non-predominant features with eligibility:** We did not observe a strong bias in the previous setting when a non-predominant, non-sensitive feature was correlated with eligibility. In this setting, we would like to evaluate whether models would exhibit any bias when such a non-predominant feature, which is correlated with eligibility, becomes a sensitive

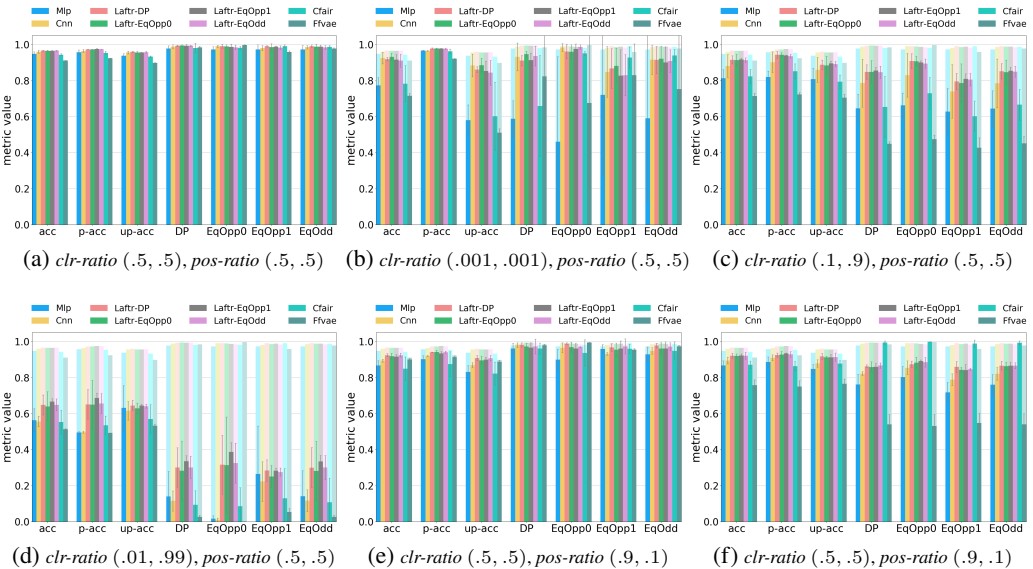

(a) *clr-ratio* (.5, .5), *pos-ratio* (.5, .5)  (b) *clr-ratio* (.001, .001), *pos-ratio* (.5, .5)  (c) *clr-ratio* (.1, .9), *pos-ratio* (.5, .5)

(d) *clr-ratio* (.01, .99), *pos-ratio* (.5, .5)  (e) *clr-ratio* (.5, .5), *pos-ratio* (.9, .1)  (f) *clr-ratio* (.5, .5), *pos-ratio* (.9, .1)

Figure 3: Comparing models on CI-MNIST dataset. 3(a) shows the balanced case. 3(b) shows the impact of Reducing unprivileged group in Setting 1. 3(c) and 3(d) show the impact of correlation of background sensitive attribute and eligibility in Setting 2. 3(e) shows the impact of correlation of non-sensitive attribute and eligibility in Setting 3. Finally, 3(f) shows the impact of correlation of small box sensitive attribute and eligibility in Setting 4. Compare all sub-figures with 3(a) to observe each biased setting's impact. In all sub-figures except 3(a), the pale colors show the decrease in performance compared to the balanced case in 3(a). Higher is better for all metrics.

attribute. Note that bias-mitigation models take sensitive attribute as input, and fairness metrics are measured for the chosen sensitive attribute.

We use the same configuration as in Setting 3, but change the sensitive attribute from background color to the small box location. Figures 3(a) and 3(f) compare the performance of different models. As we can observe, the performances of most of the models drop, except that of Cfair, which mostly maintains its previous fairness performance. The obtained results indicate that correlation of even non-predominant sensitive features with eligibility can cause bias, although the impact is lower compared to Setting 2, where the predominant component (background color) was correlated with eligibility. Hence, the level of the predominance of the sensitive attribute can change the degree of bias of the trained models. Figure 10 and Tables 46 to 53 give detailed results for all models.

**Notes on all settings.** In settings 3 and 4, we could not emulate similar scenarios on the Adults dataset, as each feature (such as age) is one out of 112 components; hence they all occupy a single dimension. Moreover, in settings 1 and 2, we could not emulate the extreme scenarios of CI-MNIST dataset either. This highlights the difficulty of emulating all scenarios in real datasets and motivates the usage of the proposed synthetic data in conjunction with real datasets.

# 5 Discussion

In this section, we discuss the results presented in section 4.

## 5.1 Model Stability and Performance

**Variation due to random seeds.** In deep learning models, the random seed is a potential source of variation due to its usage in the initialization of the model parameters and the data sampling done during training, which can converge the model to different solutions in the course of training. To assess the impact of the random seed, we trained each model three times, using three random seeds and reported the mean and standard deviation of the results – see Figures 11 and 12 in Appendix for all of the settings described in our experiments. We observed that the choice of seed could impact some models more than others. Depending on the experiment, almost all models show instability in

terms of variation of results given different seeds; however, Laftr models show more stability over all experiments. We advocate verification of any bias-mitigation method under different seeds, as this indicates how much the model is susceptible to minor training variations. This is complementary to other studies of stability, such as cross-validation studied in [36].

**Correlation between dataset features and model's prediction.** If a model's prediction is correlated with dataset features, the model could rely on correlations in the data when making predictions and thus be unfair when applied. To verify this, for each model, we measured the correlation between dataset features and fairness metrics using Spearman correlation matrices, measured separately for each setting in Section 4. The results are presented in Figure 13. We observe that all models are subject to bias as their predictions correlate with dataset features; however, Cfair is slightly less biased in capturing correlations in some setups.

One interesting observation is that despite not observing high bias in Setting 3, the correlation plot of this setting indicates that the models are still picking up on the correlation of the non-sensitive attribute and the eligibility. This suggests that better fairness results in such cases are partially due to how fairness metrics are calculated (w.r.t. the selected sensitive feature, which was the background attribute in this case), regardless of the fact that models remain biased towards unobserved yet potentially sensitive features.

**Sensitive information removal.** In bias-mitigation algorithms, the models try to remove sensitive information from the latent space. One natural question would be to what extent models are successful in achieving this goal? To verify whether the sensitive information still exists in the latent representation, we freeze a model's parameters after training and train a classifier that, given the latent features of the model, attempts to predict the sensitive attribute class. For each configuration in Section 4, we train such a classifier on its training set and report the results on the test set, which is shown in sens-acc column in Tables 6 to 53 of Settings 1 to 4.

We observe some interesting trends: First, when the sensitive attribute is predominant, there is more information about the sensitive attribute in the latent representation of models compared to the cases where the sensitive attribute is less dominant. This is observed when the background color in CI-MNIST is the sensitive attribute in settings 1, 2, and 3, compared to the small box in setting 4. On the Adult, since age is one out of 112 features, it has been relatively easy for models to remove the sensitive information. Second, despite bias-mitigation approaches exhibiting fairer results than the baselines (MLP, Cnn), in many cases, we observe that the sensitive information is higher in the latent representation of models trained with bias-mitigation algorithms. We hypothesize that the success of these bias-mitigation approaches in achieving fairer results mainly rely on the sensitive group balancing enforced through their loss functions rather than completely removing the sensitive information from their latent space.

**Model performances.** While the performance of models were often close, depending on the experiment, we often found either Cfair or Laftr to be performing slightly better compared to other models. It is worth noting that both of these models apply fairness criteria directly onto their learned latent spaces, rather than disentangling their latent representations into sensitive and non-sensitive components, as done in Ffvae. Overall, we observed that disentanglement of features was less successful in removing sensitive information. This is especially highlighted in setting 1 and 2 of the Adults dataset results – see Tables 9 and 25, respectively, and in setting 4 of the CI-MNIST dataset – see Table 49), as we observe sens-acc is higher in the non-sensitive latent space of Ffvae compared to Cfair and Laftr.

Since bias mitigation applied to a unified latent space and explicit latent space disentangling constitute two prominent bias-mitigation strategies, a natural question is whether these strategies can benefit from one another. To verify this, we merged both strategies and investigated the effect on the models' performances. In one setup, we merged Laftr-DP with Ffvae, and in another setup, we merged Cfair with Ffvae. Check Section E.8 for details of merging models. We evaluated these merged models in settings 1 and 2 of the Adults dataset. Tables 56 to 59 present results for the merged models. We observe an improvement of the fairness metrics (DP, EqOpp, EqOdd) in joint models compared to individual models while accuracy has not changed or deteriorated. In particular, when comparing the results in Tables 56 to 59 to the tables obtained from individual models, we observe on average over different cases the fairness metrics DP, EqOpp0, EqOpp1, and EqOdd have improved in Ffvae+Cfair by 5.83%, 3.41%, 9.15%, 5.98% and in Ffvae+Laftr by 9.63%, 3.4%, 19.4%, 10.53%, while accuracy has improved in Ffvae+Cfair by only 1.15% and dropped in Ffvae+Laftr by 12.66%. This emphasizes that further investigation in merging the seemingly different bias-mitigation strategies can bring improvement *w.r.t* fairness metrics while maintaining model accuracy.

## 5.2 Sources of Bias

In our experiments, in addition to random seed we have considered three different sources of bias:

**Reduced representation of the unprivileged group.** This scenario causes bias due to the small ratio of the unprivileged group compared to the privileged group. In this case, the bias can be two-fold; one is due to the imbalance between the two groups, another can be due to scarcity of data from the unprivileged group. In particular, the *clr-ratios* of $(0.1, 0.1)$, $(0.01, 0.01)$, and $(0.001, 0.001)$ are all subject to group imbalances. However, the latter two cases have 250 and 25 samples from the unprivileged group in CI-MNIST , compared to the 24,750 and 24,975 samples respectively from the privileged group. So, the model not only has to handle the group imbalance but also the secondary aspect of the bias, which is scarcity of the data.

To further disentangle the cause of bias, we performed an experiment where we kept the ratio of $(0.001, 0.001)$ in all settings, but increased the number of total samples by 10, 100, and 1000 times in order to alleviate data scarceness, especially in the $1000\times$ case where the under-represented group is as numerous as the original dataset size. The results are shown in Tables 54 for Cnn (without any bias-mitigation strategy) and in 55 for Laftr-EqOpp0, which is one of the strongest bias-mitigation models. Interestingly, in these models, the source of bias is different. In Laftr, the scarceness of data seems to be a source of bias, and as the number of samples in the under-represented group increases, the model's performance improves, although it is still not as good as in the balanced setup shown in the first row of Table 20. In Cnn, however, the imbalance in the data distribution seems to be the main cause of bias, as the model does not effectively leverage the increasing number of samples. These results indicate that different models may be susceptible to different sources of bias, including both data imbalance and data scarcity.

**Correlation of a feature with eligibility.** In some datasets, the unprivileged group, might exhibit lower eligibility, *e.g.,* due to historical reasons like lack of access to facilities or limited resources compared to a privileged group [42]. As observed in settings 2 and 4, correlations between features and eligibility in the data can be somewhat easily picked up by the model, causing bias at test time. In such settings, however, we observe that if the sensitive attribute/features are predominant, i.e., occupies a more significant portion of the input data, the bias becomes stronger, causing the fairness metrics to drop further, as observed when comparing setting 2 to setting 4. Moreover, in both settings we observe that as the correlation increases, the degree of bias in models augments.

**Impact of non-predominant correlated features.** The bias-mitigation algorithms work based on the assumption that the sensitive or biased attributes are known *a priori*, and hence can be addressed by removing these features from the learned representations. However, some features, especially small (non-predominant) ones, might not be noticed by the model or a potential annotator. For example, wearing glasses can correlate with age. If this is not known *a priori*, the bias-mitigation algorithm will most likely not address it adequately. Our results in setting 3 and the Spearman correlation plot in this setting indicate that such non-predominant but correlated features can be a source of bias and the models still capture the correlation in the data.

## 6 Conclusion

With the increasing use of representation learning algorithms in automatic decision-making tools [43, 44, 45, 46], the rigorous benchmarking of bias-mitigation algorithms has become imperative. In this paper, we introduced a framework to benchmark bias-mitigation algorithms in a wide variety of scenarios to push these methods to their limits. We systematically compared and analyzed a set of deep learning-based bias-mitigation models using the proposed benchmark on challenging variations of the introduced synthetic CI-MNIST and Adults datasets, where we control the correlation and balance between different dataset subgroups. Although the models under analysis proved their effectiveness to the community and were able to perform on some of our settings and dataset variants adequately, we showed that we could purposefully push them to their breaking point. Throughout our analysis, we observed that these models could exploit and hence induce biases when: 1) the unprivileged group is under-represented, which can be due to either imbalance or scarceness of its population; and 2) there is a correlation between the sensitive attribute and the eligibility. In both cases, we observed a stronger bias when imbalance or correlation increases. Our results also showed

that the dimensionality/predominance of the biased input features could affect the model's degree of bias. Further, we found that the sensitive information is present in the latent representation of the models trained with bias-mitigation strategies, and hence using representation from these models for downstream tasks can still result in a biased treatment. Our results also showed that some models are more susceptible to the random seed than others and hence variation of random seed can be a source of bias. Therefore, not all models may be well-suited to operate in all scenarios, and using models without a proper understanding of their limitations could lead to undesirable consequences. We also note that with the abundance of fairness metrics there is room for improvement in model selection, as we cannot pinpoint a single model that works best across all metrics. We hope our benchmark serves as a starting point to verify robustness of bias-mitigation models. Finally, we will release our dataset and codebase to the community to provide a framework for evaluating deep learning-based bias-mitigation algorithms.

## Acknowledgement

We would like to thank Benjamin Fish for reading the manuscript in multiple occasions and providing insightful inputs. We also thank Elliot Creager for providing valuable inputs in modelling fairness algorithms. Finally, we are thankful to Fernando Diaz and Philip Bachman for their feedback in this project.

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

# A  Paper Checklist

## A.1  Potential negative societal impacts

While we report results using the best currently established fairness metrics, there is neither a universal fairness metric nor a universally accepted definition of fairness. Therefore, although we take steps towards proper benchmarking of bias mitigation algorithms, one should still be cautious about the chosen metrics before using them in real-world applications. Group parity metrics like DP, EqOpp0, EqOpp1, EqOdd can suffer from statistical limitations, as the true underlying protected group distributions might differ regardless of other unprotected features used for prediction, thus enforcing these metrics can lower accuracies and can harm the very groups designed to protect [47].

Hence before using any specific metric, the true data distribution should be studied well by designing suitable interventions. Real-world assessments and understanding consequences also help in achieving equitable metrics. Today with representation learning methods used in automatic decision-making applications [43, 44, 45, 46], more interpretable and explainable models are necessary to avoid any harmful consequences.

## A.2  Limitations

The results presented in this work are obtained using binary sensitive attributes. It would be interesting to extend the present work to non-binary sensitive attributes in addition to observing the impact of bias-mitigation methods on multiple sensitive features. Due to the extensiveness of evaluating all these cases, we focused only on single binary attributes. Also, just mitigating the effect of sensitive features might not be enough, as there can be proxy features present, which are partially correlated with the sensitive features [47], it is aspiring to see fairness research that exploits the correlations between input features.

On a separate note, we have evaluated some of the recent promising bias-mitigation algorithms out of many proposed models. This field is expanding rapidly, and we could not evaluate all possible models. It would be interesting to evaluate other promising models in the proposed setups of this paper and try the proposed settings on more datasets to observe any potential change of performance due to difference in the modality of data.

## A.3  Privacy and author consent

**CI-MNIST** : This data is an extension of the publicly available MNIST dataset [39], which does not contain any personal data. Yann LeCun and Corinna Cortes hold the copyright of MNIST dataset, which is a derivative work from original NIST datasets. MNIST dataset is made available under the terms of the Creative Commons Attribution-Share Alike 3.0 license (CC BY-SA 3.0).

**Adult**: The Adult dataset was originally extracted by Barry Becker from the 1994 Census bureau database and the data was first cited in [48]. It was donated by Ronny Kohavi and Barry Becker (Data Mining and Visualization, Silicon Graphics) and publicly released to the community on the UCI Data Repository [10]. It is licensed under Creative Commons Public Domain (CC0).

We do not put these datasets in our repository; instead, we provide code and guidelines on processing the original dataset to obtain the dataset variants used in our experiments.

## A.4  Reproducibility

We have released our code at https://github.com/charan223/FairDeepLearning. We have provided the instructions to reproduce all experiments reported in the paper. Model details are provided in Section C. Dataset, architectural, and hyper-parameter details are presented in Section D. The detailed results on all experiments are presented in Section E. For each chart, we provide confidence-interval around the mean and run each experiment with three different seeds. We trained about 3000 models on 2 16GB NVIDIA Tesla V100 GPUs for 14 days.

# B  Fairness metrics

Table 1 presents all fairness metrics and their mathematical formulations that we use in our evaluations. In Table 2, we present the comprehensive list of fairness metrics taken from the literature along with their mathematical definitions and abbreviations. In our code-base that we will release to the fairness community, all these metrics are provided and can be used for evaluation. The provided tool by [36] is used to compute all of the metrics.

Table 1: $X, Y, S$ denote the input, label, and the sensitive attribute. $\hat{Y}$ and $p$ are the model's prediction and the output probability of the model. For all metrics, 1 indicates the perfect and 0 the lowest value.

| Fairness Criteria | Formulation | Short form |
|---|---|---|
| Demographic Parity | $1 - \|p(\hat{Y} = 1\|S = \text{Protected}) - p(\hat{Y} = 1\|S = \text{Unprotected})\|$ | DP |
| Equality of Opportunity (w.r.t $y = 1$) | $1 - \|p(\hat{Y} = 1\|Y = 1, S = \text{Unprotected}) - p(\hat{Y} = 1\|Y = 1, S = \text{Protected})\|$ | EqOpp1 |
| Equality of Opportunity (w.r.t $y = 0$) | $1 - \|p(\hat{Y} = 1\|Y = 0, S = \text{Unprotected}) - p(\hat{Y} = 1\|Y = 0, S = \text{Protected})\|$ | EqOpp0 |
| Equality of Odds | $0.5 \times [\text{EqOpp0} + \text{EqOpp1}]$ | EqOdd |
| unprotected-accuracy | $p(\hat{Y} = y\|Y = y, S = \text{Unprotected})$ | up-acc |
| protected-accuracy | $p(\hat{Y} = y\|Y = y, S = \text{Protected})$ | p-acc |
| accuracy | $0.5 \times [\text{up-acc} + \text{p-acc}]$ | acc |

# C  Models

In this section, we describe in detail the models used in our evaluations.

**Baseline Model.** We use Mlp and Cnn as our baseline models, which given an input image $x$, predicts the probability $p$ of the eligibility criteria. This probability is then transformed into a classification prediction $\hat{y}$. These models do not leverage any bias mitigation algorithm and are trained using a standard cross-entropy loss. It is meant to show how fair a baseline deep learning model would perform under different fairness criteria.

**Learning Adversarially Fair and Transferable Representations (Laftr).** Laftr [6] is an adversarial based bias mitigation algorithm within the scope of representation learning. In the supervised version of Laftr, given an input $x$, it first learns a latent encoded representation $z$ that is passed to the discriminator to be debiased. The learned representation is then passed to a classifier to predict the task of interest $y$. The discriminator is trained by minimizing

$$\mathcal{L}_{fair}^{Laftr} = \mathbb{E}_{x,y,s \in \mathcal{D}} \mathcal{L}_{\mathcal{S}}(D(z, y), s) \tag{1}$$

where $\mathcal{L}_{\mathcal{S}}$ is the adversarial loss, and $y$ is only passed in debiasing models aimed for *equality of odds* and *equality of opportunity* fairness metrics. The encoder and classifier are trained jointly by minimizing

$$\mathcal{L}_{Laftr} = \mathbb{E}_{x,y,s \in \mathcal{D}} \mathcal{L}_{\mathcal{Y}}(C(z), y) - \gamma \mathcal{L}_{fair}^{Laftr} \tag{2}$$

where $z = E(x)$ is the encoded feature, passed to both the classifier $C$ and the discriminator $D$. The first term on the right side of the equation measures the classification loss (denoted as $\mathcal{L}_{cl}^{Laftr}$), and the second term (or the fairness objective) gets the adversarial gradients from the discriminator regarding the sensitive attribute $s$. Following the original paper, four variants of Laftr model are considered that represent the desired fairness criteria via $\mathcal{L}_{fair}^{Laftr}$. This includes:
(i) Laftr-DP in which the fairness objective is defined as

$$\mathcal{L}_{DP}^{Laftr} = 1 - \sum_{s \in \{0,1\}} \mathbb{E}_{x,s \in \mathcal{D}_s} |D(z) - s| \tag{3}$$

(ii) Laftr-EqOpp0 in which the fairness objective is considered as

$$\mathcal{L}_{EqOpp0}^{Laftr} = 1 - \sum_{s \in \{0,1\}, y=0} \mathbb{E}_{x,s \in \mathcal{D}_s^y} |D(z) - s| \tag{4}$$

(iii) Laftr-EqOpp1 whose fairness objective $\mathcal{L}_{EqOpp1}^{Laftr}$ is obtained by replacing $y = 1$ in Eq. (4)
(iv) Laftr-EqOdd with the equality of odds fairness objective denoted as $\mathcal{L}_{EqOdd}^{Laftr}$ which is the sum of $\mathcal{L}_{EqOpp0}^{Laftr}$ and $\mathcal{L}_{EqOpp1}^{Laftr}$.

Table 2: Fairness metrics. $X, Y, S$ denote respectively the input sample, the ground truth label, and the sensitive attribute. $p$ is the output probability of the model and $\hat{Y}$ is the model's prediction. For the metrics presented in this table, the sensitive attribute $S$ takes binary values in $\{0, 1\}$.

| Fairness Criteria | Definition | Abbreviation |
|---|---|---|
| Group conditioned s-accuracy | $p(\hat{Y} = y \mid Y = y, S = s)$ | $s$-accuracy |
| $s$-True positive [37] | $\lvert \{x \mid \hat{y} = 1 \text{ for } (x, y = 1, S = s) \in X\} \rvert$ 
 where $\lvert \cdot \rvert$ refers to cardinality of a set | $s$-TP |
| $s$-False positive [37] | $\lvert \{x \mid \hat{y} = 1 \text{ for } (x, y = 0, S = s) \in X\} \rvert$ 
 where $\lvert \cdot \rvert$ refers to cardinality of a set | $s$-FP |
| $s$-False negative [37] | $\lvert \{x \mid \hat{y} = 0 \text{ for } (x, y = 1, S = s) \in X\} \rvert$ 
 where $\lvert \cdot \rvert$ refers to cardinality of a set | $s$-FN |
| $s$-True negative [37] | $\lvert \{x \mid \hat{y} = 0 \text{ for } (x, y = 0, S = s) \in X\} \rvert$ 
 where $\lvert \cdot \rvert$ refers to cardinality of a set | $s$-TN |
| $s$-True positive rate [36] 
 = $s$-positive predictive value ($s$-PPV) | $p(\hat{Y} = 1 \mid Y = 1, S = s)$ | $s$-TPR |
| $s$-True negative rate | $p(\hat{Y} = 0 \mid Y = 0, S = s)$ | $s$-TNR |
| $s$-False positive rate | $p(\hat{Y} = 1 \mid Y = 0, S = s)$ 
 equivalent to $1 - s$-TNR | $s$-FPR |
| $s$-False negative rate | $p(\hat{Y} = 0 \mid Y = 1, S = s)$ 
 equivalent to $1 - s$-TPR | $s$-FNR |
| $s$-Balanced classification rate | $0.5 \times [p(\hat{Y} = 1 \mid Y = 1, S = s) + p(\hat{Y} = 0 \mid Y = 0, S = s)]$ 
 equivalent to $0.5 \times (s$-TPR + $s$-TNR $)$ | $s$-BCR |
| Equality of odds [49] & [27] 
 = Equalized odds [49] 
 = conditional procedure accuracy equality [50] 
 = disparate mistreatment [51] | $p(\hat{Y} = \hat{y} \mid Y = y) = p(\hat{Y} = \hat{y} \mid Y = y, S = s)$ 
 equivalent to $[p(\hat{Y} = 1 \mid Y = 1, S = 1) = p(\hat{Y} = 1 \mid Y = 1, S = 0)$ and 
 $p(\hat{Y} = 1 \mid Y = 0, S = 1) = p(\hat{Y} = 1 \mid Y = 0, S = 0)]$ 

 equivalent to [ 1-TPR = 0-TPR and 0-TNR = 1-TNR ] | - |
| $s$-calibration+ [36] | $p(Y = 1 \mid \hat{Y} = 1, S = s)$ | - |
| $s$-calibration− [36] | $p(Y = 1 \mid \hat{Y} = 0, S = s)$ | - |
| Conditional use accuracy equality [50] | $[p(Y = 1 \mid \hat{Y} = 1, S = 1) = p(Y = 1 \mid \hat{Y} = 1, S = 0)$ and 

 $p(Y = 0 \mid \hat{Y} = 0, S = 1) = p(Y = 0 \mid \hat{Y} = 0, S = 0)]$ 
 equivalent to [0-calibration+ = 1-calibration+ and 0-calibration− = 1-calibration−] | - |
| Calders and Verwer [52] | $1 - [p(\hat{Y} = 1 \mid S = 1) - p(\hat{Y} = 1 \mid S \neq 1)]$ | CV |
| Demographic parity [49] & [27] 
 = Group fairness [17] 
 = statistical parity [17] 
 = equal acceptance rate [53] | $p(\hat{Y}) = p(\hat{Y} \mid S)$ 
 equivalent to $1 -$ CV 
 equivalent to $p(\hat{Y} = 1 \mid S = 1) = p(\hat{Y} = 1 \mid S \neq 1)$ | DP |
| Disparate Impact [54] & [40] | $\frac{p(\hat{Y} = 1 \mid S \neq 1)}{p(\hat{Y} = 1 \mid S = 1)}$ | DI |
| Equality of opportunity with respect to $y$ [49] | $p(\hat{Y} = \hat{y} \mid Y = y) = p(\hat{Y} = \hat{y} \mid Y = y, S = s)$ 
 Equality of odds is stronger than equality of opportunity | - |
| False positive error rate balance [55] 
 = predictive equality [56] | $p(\hat{Y} = 1 \mid Y = 0, S = 1) = p(\hat{Y} = 1 \mid Y = 0, S = 0)$ 
 equivalent to $p(\hat{Y} = 0 \mid Y = 0, S = 1) = p(\hat{Y} = 0 \mid Y = 0, S = 0)$ 
 equivalent to 1-TNR = 0-TNR 
 equivalent to [Equality of opportunity with respect to $y = 0$] | - |
| False negative error rate balance [55] 
 = equal opportunity [16] & [49] | $p(\hat{Y} = 0 \mid Y = 1, S = 1) = p(\hat{Y} = 0 \mid Y = 1, S = 0)$ 
 equivalent to $p(\hat{Y} = 1 \mid Y = 1, S = 1) = p(\hat{Y} = 1 \mid Y = 1, S = 0)$ 
 equivalent to 1-TPR = 0-TPR 
 equivalent to [Equality of opportunity with respect to $y = 1$] | - |
| Matthews correlation coefficient | $\frac{\text{TP} \times \text{TN} - \text{FP} \times \text{FN}}{\sqrt{(\text{TP} + \text{FP})(\text{TP} + \text{FN})(\text{TN} + \text{FP})(\text{TN} + \text{FN})}}$ | MCC |

**Conditional Learning of Fair Representations (Cfair).** Proposed by [7], this model leverages two adversarial networks $h_0$ and $h_1$, predicting sensitive attribute $s$ respectively for class labels $Y = 0$ and $Y = 1$. Cfair depends on an objective function called the balanced error rate (BER) [54, 57], which guarantees small joint error across demographic groups. The BER represents the sum of false positive rate and false negative rate. Therefore, it is equal to minimizing the below two conditional errors. $\text{BER}_{\mathcal{D}}(\hat{Y} \| Y)$ is defined as

$$\text{BER}_{\mathcal{D}}(\hat{Y} \| Y) \propto p(\hat{Y} = 1 \mid Y = 0) + p(\hat{Y} = 0 \mid Y = 1). \tag{5}$$

and $\text{BER}_{\mathcal{D}}(\hat{S} \| S)$ is defined similarly, where $\hat{S}$ is the predicted sensitive random variable. Cfair is optimized based on the following min-max formulation.

$$\mathcal{L}_{Cfair} = \min_{C,E} \max_{h_0, h_1} \left( \text{BER}_{\mathcal{D}}(C(E(X)) \| Y) - \gamma \mathcal{L}_{DP}^{Cfair} \right) \tag{6}$$

where

$$\mathcal{L}_{DP}^{Cfair} = \text{BER}_{\mathcal{D}^{y=0}} (h_0(E(X)) \| S) + \text{BER}_{\mathcal{D}^{y=1}} (h_1(E(X)) \| S) \tag{7}$$

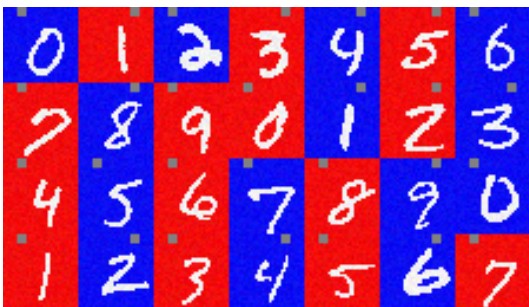

Figure 4: Sampled images from our dataset.

This approach proposes that using the balanced error rate along with the conditional alignment helps in achieving equalized odds across the groups without impacting demographic parity.

Cfair-EO is a variant of the Cfair model, which considers Cross-Entropy loss instead of BER loss for the classifier $C$ to achieve equalized odds. In case of equal target class distribution, Cfair and Cfair-EO are the same, refer to Eq. (9) in the Appendix.

**Flexibly Fair Representation Learning by Disentanglement (Ffvae).** Inspired by FactorVAE [29], Ffvae [8] performs disentanglement by factorizing latent space. It learns a disentangled representation of the inputs, which is flexibly fair because it can be easily modified at test time to achieve demographic parity across various groups.

Given $x$, $s = (s_1, \ldots, s_N)$, $b = (b_1, \ldots, b_N)$, and $z$ being respectively the input, the sensitive attribute, the sensitive latent, and non-sensitive latent, with $N$ indicating the number of sensitive or non-sensitive features (depending on the dataset), Ffvae trains an encoder $q(z, b|x)$, a decoder $p(x|z, b)$, as well as an adversarial network. The latent representation is disentangled into sensitive $b$ and non sensitive $z$ latent attributes by encouraging both $\mathrm{MI}(b, z)$ and $\mathrm{MI}\,(b_i, s_j)\,, \forall i \neq j$ to be low, where MI represents mutual information. Ffvae objective is defined as

$$\mathcal{L}_{\mathrm{Ffvae}}(p, q) = \mathbb{E}_{q(z,b|x)}[\log p(x \mid z, b) + \alpha \log p(s \mid b)]] - \gamma D_{KL}(q(z,b)\|q(z) \prod_j q\,(b_j))$$
$$- D_{KL}(q(z, b \mid x)\|p(z, b))$$
(8)

Eq.(8) has two terms, the first term consists of a reconstruction term (on left) and a *predictiveness* term $p(s \mid b)$, which aligns sensitive attributes to its respective sensitive latents, the second term is the *disentanglement* term which decorrelates the sensitive latent representation $b$ from $z$ using an adversarial network.

In addition to the models mentioned above, we also did experiments with [9] (based on the code released by authors), however, the model was very unstable on our dataset configurations even after extensive hyper-parameter search. We hypothesize that this is due to applying adversarial training directly to the class labels, which makes the model unstable, as indicated by the authors. Due to unstable results, we dropped this model from our evaluation.

## D  Experimental Setup

*CI-MNIST dataset:* Figure 4 shows samples of the dataset used in our experiments. Unless otherwise stated, we used 50000 images for the training set, 10000 images for each validation and test sets. In CI-MNIST experiments, the eligible and ineligible groups represent each 50% of the training data in both train and test sets. However, while the train set can be imbalanced with respect to sensitive attributes, the test set is always balanced. We initially pad the input image of size 28x28 on the top and sides to give a 32x32 image for the dataset creation. Blue and red colors with a 10% gaussian noise are used as background colors. For the small box, we used a 4x4 sized gray-colored box in the center of the top-left half or top-right half of the padded region of the image. We have experimented with multiple background colors, box colors and box sizes to understand the impact of colors, positions, sizes of the features on our models. Our motivation in choosing the current features is that the models

can easily notice these features but find them difficult to remove. CI-MNIST currently supports multiple sensitive features (multiple background colors and positions of boxes). For more details refer to the released codebase.

*Adult dataset:* For training on the Adult dataset, we used the full Adult training set consisting of a total of 30,162 samples. We used 20% of the training set as the validation set. In the Adult dataset, the eligible and ineligible groups represent each 25%, 75% of the training data; hence the data is imbalanced with respect to target label with a skew towards ineligible lower-income class (<=50k). We use age as sensitive attribute and threshold it in the training sets (as indicated in Table 3) to create various *age-ratios*. Note that in all *age-ratios* the size of the training dataset does not change, and the eligible and ineligible groups remain at 25% and 75%. However, through thresholding of age as the sensitive attribute, we change the number of people in privileged and unprivileged groups. We find age-thresholds that closely resemble the dataset ratios used in CI-MNIST , in terms of the percentage of unprivileged individuals in eligible and ineligible groups.

We change the original test set to create a balanced version of it for testing, meaning it is balanced in terms of both sensitive attributes and target classes. Hence, the number of testing samples vary for each *age-ratio*. This is because we use age for the sensitive attribute, and when we change its threshold, the number of examples belonging to unprivileged and privileged group changes. We drop the minimum number of samples from the bigger subgroup of sensitive attribute and target class to make the dataset balanced. In Table 3, we mention the age thresholds used for the unprivileged group to achieve our desired *age-ratios* and also indicate the test-set size in each case. The remaining ages in either eligible and ineligible groups are considered privileged.

| age-ratio | unprivileged age threshold | test set size |
|---|---|---|
| (0.5, 0.5) | 25 <= age < 44 | 7336 |
| (0.1, 0.1) | 32 <= age < 36 | 1708 |
| (0.01, 0.01) | 71 <= age < 75 | 208 |
| (0.66, 0.33) | 38 <= age < 60 | 5480 |
| (0.06, 0.36) | 0 <= age < 30 | 908 |

Table 3: Thresholds of age and test set size used for various *age-ratios*. The test set is balanced in terms of both sensitive attribute and target class, while the train set is imbalanced and is of fixed size 30,162.

*Architecture:* In all models, we used three fully connected layers for discriminator and classifier networks. In the encoder network, we used three fully connected layers for all models except Ffvae, in which we used a convolutional encoder and decoder networks for increased training stability [29]. Leaky ReLU is used for all activation functions, and Glorot [58] is used to initialize all weights. The models are trained using Adam optimizer with a learning rate of 1e-3. Models are trained for 500 epochs, with early stopping of 5 epochs patience on the validation set's loss to find the best model.

*Baseline Mlp Setup:* The baseline Mlp model consists of an encoder for the input image and a classifier for eligibility prediction. Cross entropy loss is used for optimization.

*Baseline Cnn Setup:* The baseline Cnn model consists of an Cnn encoder for the input image and an Mlp classifier for eligibility prediction. Cross entropy loss is used for optimization.

*Laftr Setup:* The model consists of an encoder, a classifier, and a discriminator. We used an adapted PyTorch version of the original codebase released by the authors of the original paper [6]. Following the original code's training method, we train the encoder, classifier and train the discriminator in alternate steps. We used two discriminator iterations per encoder-classifier iteration and applied cross-entropy loss for optimization of both the classifier and discriminator. We used the default classification coefficient of 1.0 and used five values of adversarial coefficient $\gamma \in [0.1, 0.5, 1, 2, 4]$, as proposed in the original paper.

*Cfair Setup:* The model consists of an encoder, a classifier, and two discriminators (one for each eligibility class label). We used the code provided by the authors to run the experiments. We experimented with five values of adversarial coefficient $\gamma \in [0.1, 1, 10, 100, 1000]$, as proposed in the original paper. The binary loss (0-1 loss) in Eq.6 is NP-hard to optimize directly [59, 60], hence the model uses a convex relaxation of the binary loss, which is a weighted cross-entropy loss as shown

below.

$$\mathcal{D}(\widehat{Y} \neq y \mid Y = y) = \frac{\mathcal{D}(\widehat{Y} \neq y, Y = y)}{\mathcal{D}(Y = y)}$$
$$\leq \frac{\mathrm{CE}_{\mathcal{D}^y}(\widehat{Y} \| Y)}{\mathcal{D}(Y = y)} \tag{9}$$

*Ffvae Setup:* The model consists of a convolutional encoder, a convolutional decoder, a fully connected classifier, and a fully connected discriminator. We used the code provided by authors to run the experiments. We applied adversarial coefficient $\gamma \in [10, 50, 100]$ and the alignment coefficient $\alpha \in [10, 100, 1000]$. We observed that the training of Ffvae becomes unstable for higher values of $\gamma$. This is due to the fact that the stability between *predictiveness* and *disentanglement* gets harder to achieve as they work against each other when the sensitive attribute and the eligibility are correlated. Ffvae model takes ELBO loss for the VAE and approximates the *disentanglement* term using the mean error difference between discriminator logits [29]. The model uses cross-entropy loss for the *predictiveness* term and the discriminator network.

In CI-MNIST experiments, we kept the widths of encoder, decoder, discriminator constant at 32, and the encoded latent representation size is 16 for all models. We experimented with two values of classifier widths 32, 64 and were unable to observe the trend which is recently emphasized by [61] that increasing model capacities may lead to being unfair toward minorities while accuracy is getting better. However, this needs to be further investigated. Tables 4, 5 show the architectural details for CI-MNIST and Adult experiments.

Table 4: Architectures used for Baseline Mlp, Baseline Cnn, Laftr, Cfair, Ffvae models for CI-MNIST dataset.

| Mlp Encoder | Mlp Classifier/Discriminator | Cnn Encoder |
|---|---|---|
| Input $\in \mathbb{R}^{3072}$ | Input $\in \mathbb{R}^{16}$ | Input $32 \times 32 \times 3$ image |
| FC. 32 LReLU | FC. 32 LReLU | $4 \times 4$ conv. 32 LReLU. stride 2, padding 1 |
| FC. 32 LReLU | FC. 32 LReLU | $4 \times 4$ conv. 64 LReLU. stride 2, padding 1 |
| FC. 16 LReLU | FC. 2 LReLU | $4 \times 4$ conv. 64 LReLU. stride 2, padding 1 |
| | | $4 \times 4$ conv. 256 LReLU. stride 1 |
| | | $1 \times 1$ conv. 16 LReLU. |

| Ffvae Encoder | | Ffvae Decoder |
|---|---|---|
| Input: $32 \times 32 \times 3$ image | | Input $\in \mathbb{R}^{16}$ |
| $4 \times 4$ conv. 32 LReLU. stride 2, padding 1 | | FC. 128 LReLU |
| $4 \times 4$ conv. 64 LReLU. stride 2, padding 1 | | FC. 1024 LReLU, Resize $64 \times 4 \times 4$ |
| $4 \times 4$ conv. 64 LReLU. stride 2, padding 1 | | $4 \times 4$ upconv. 64 LReLU. stride 2, padding 1 |
| Flatten 1024, FC. 128 LRELU | | $4 \times 4$ upconv. 32 LReLU. stride 2, padding 1 |
| FC. $2 \times 16$ | | $4 \times 4$ upconv. 3 LReLU. stride 2, padding 1 |

In Adult experiments, we kept the widths, latent representation sizes the same as their respective original papers.

Table 5: Architectures used for Baseline Mlp, Laftr, Cfair, Ffvae models for Adult dataset.

| Mlp Encoder | Mlp Classifier | Ffvae Encoder | Ffvae Classifier/ Discriminator |
|---|---|---|---|
| Input $\in \mathbb{R}^{112}$ | Input $\in \mathbb{R}^{16}$ | Input $\in \mathbb{R}^{112}$ | Input $\in \mathbb{R}^{60}$ |
| FC. 32 LReLU | FC. 32 LReLU | FC. 200 LReLU | FC. 200 LReLU |
| FC. 32 LReLU | FC. 32 LReLU | FC. 60 LReLU | FC. 2 LReLU |
| FC. 16 LReLU | FC. 2 LReLU | | |
| Laftr Encoder | Laftr Classifier/Discriminator | Cfair Encoder | Cfair Classifier/Discriminator |
| Input $\in \mathbb{R}^{112}$ | Input $\in \mathbb{R}^{8}$ | Input $\in \mathbb{R}^{112}$ | Input $\in \mathbb{R}^{60}$ |
| FC. 8 LReLU | FC. 2 LReLU | FC. 60 LReLU | FC. 0/50 LReLU |
| | | | FC. 2 LReLU |

For sensitive information removal experiments in Section 5.1, sensitive features are predicted from latent representations from model-specific encoders. We use the same architecture as Mlp Classifier in Table 5 for these experiments.

*Hyperparameter details*:

**Laftr.** We use adversarial coefficient $\gamma \in [0.1, 0.5, 1, 2, 4]$ as hyperparameter as proposed in the original paper and we use two discriminator iterations per encoder-classifier iteration.

**Cfair.** We use adversarial coefficient $\gamma \in [0.1, 1, 10, 100, 1000]$ as hyperparameter as proposed in the original paper.

**Ffvae.** We use adversarial coefficient $\gamma \in [10, 50, 100]$ and the alignment coefficient $\alpha \in [10, 100, 1000]$ as hyperparameters as proposed in the original paper. We also use patience epochs 5 for early stopping in VAE training as a hyperparameter.

Other general hyperparameters considered for all the models include classifier, encoder, and discriminator widths, number of layers, and latent representation size with values mentioned in Tables 4 and 5. We take 5 epochs as stopping patience, use Adam as an optimizer with a learning rate of 1e-3. Please check our repository for a complete set of hyper-parameters and training setups.

# E   Experiments and Results

## E.1   Impact of reducing representation of unprivileged group

We report the complete set of results for debiasing models of Mlp, Cfair, Ffvae, Laftr-EqOdd, Laftr-EqOpp1, Laftr-EqOpp0, and Laftr-DP, in Tables 6 to 21, corresponding to the experimental setup described in Setting 1 of Section 4 in the main paper. Each pair in *clr-ratio* column indicate $(b_e, b_o)$, which is the ratio of images with blue background for (even=eligible, odd=ineligible) data. Figures 5, 6 compare all models side-by-side. Note that to report results, we initially averaged metrics over three seeds, then for each metric, the best value over the fairness coefficients of the models is reported on the test set.

Table 6: Mlp results when decreasing minority representation for Adult dataset, sensitive attribute:$age$, selected best result per attribute

| (u-elg, u-inelg) | acc | p-acc | up-acc | DP | EqOpp0 | EqOpp1 | EqOdd | sens-acc |
|---|---|---|---|---|---|---|---|---|
| (0.5, 0.5) | 0.78 | 0.8 | 0.75 | 0.97 | 0.99 | 0.92 | 0.96 | 0.66 |
| (0.1, 0.1) | 0.74 | 0.77 | 0.71 | 0.96 | 0.99 | 0.9 | 0.95 | 0.51 |
| (0.01, 0.01) | 0.74 | 0.81 | 0.66 | 0.91 | 0.97 | 0.77 | 0.87 | 0.54 |

Table 7: Cfair results when decreasing minority representation for Adult dataset, sensitive attribute:$age$, selected best result per attribute

| (u-elg, u-inelg) | acc | p-acc | up-acc | DP | EqOpp0 | EqOpp1 | EqOdd | sens-acc |
|---|---|---|---|---|---|---|---|---|
| (0.5, 0.5) | 0.82 | 0.84 | 0.81 | 0.99 | 0.97 | 0.99 | 0.98 | 0.54 |
| (0.1, 0.1) | 0.8 | 0.82 | 0.79 | 0.96 | 0.99 | 0.98 | 0.98 | 0.51 |
| (0.01, 0.01) | 0.8 | 0.87 | 0.73 | 0.85 | 0.92 | 0.68 | 0.8 | 0.54 |

Table 8: Cfair-EO results when decreasing minority representation for Adult dataset, sensitive attribute:$age$, selected best result per attribute

| (u-elg, u-inelg) | acc | p-acc | up-acc | DP | EqOpp0 | EqOpp1 | EqOdd | sens-acc |
|---|---|---|---|---|---|---|---|---|
| (0.5, 0.5) | 0.78 | 0.8 | 0.75 | 0.99 | 0.99 | 0.98 | 0.98 | 0.52 |
| (0.1, 0.1) | 0.73 | 0.77 | 0.69 | 0.98 | 0.99 | 0.96 | 0.97 | 0.51 |
| (0.01, 0.01) | 0.74 | 0.8 | 0.67 | 0.96 | 0.99 | 0.88 | 0.94 | 0.54 |

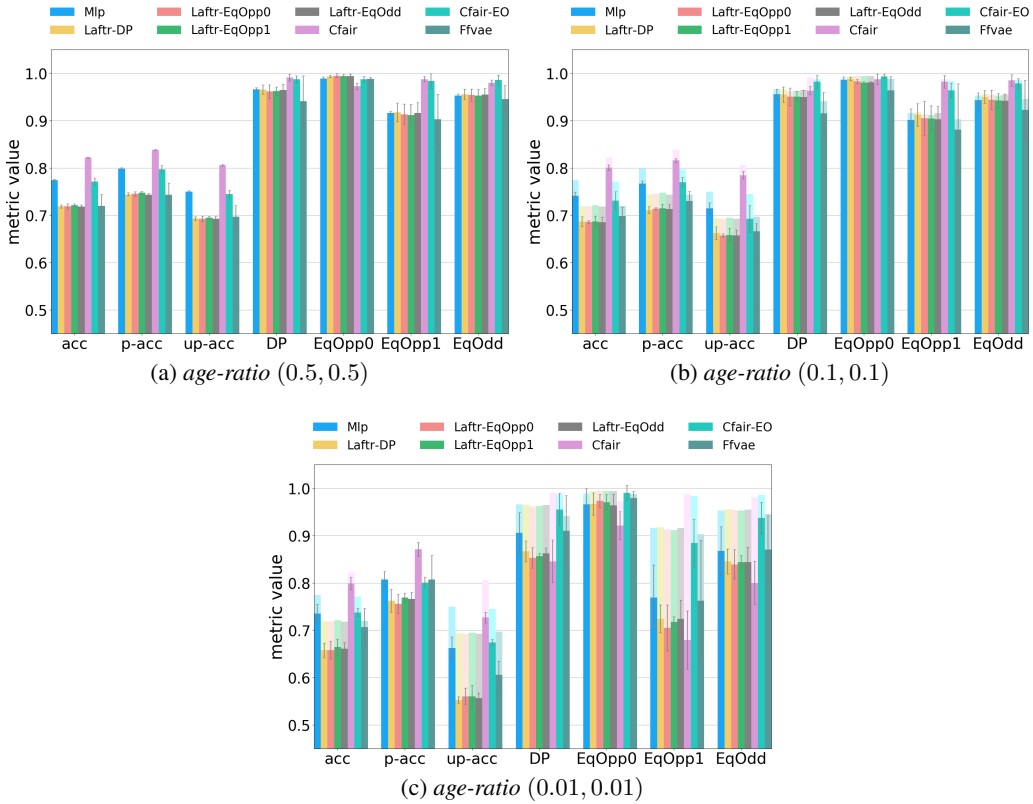

Figure 5: Comparing different models while decreasing minority representation for Adult dataset. In sub-figures 5(b) and 5(c) the pale colors show the decrease in performance compared to the balanced case in 5(a).

Table 9: Ffvae results when decreasing minority representation for Adult dataset, sensitive attribute:$age$, selected best result per attribute

| (u-elg, u-inelg) | acc | p-acc | up-acc | DP | EqOpp0 | EqOpp1 | EqOdd | sens-acc |
|---|---|---|---|---|---|---|---|---|
| (0.5, 0.5) | 0.72 | 0.74 | 0.7 | 0.94 | 0.99 | 0.9 | 0.95 | 0.93 |
| (0.1, 0.1) | 0.7 | 0.73 | 0.67 | 0.92 | 0.96 | 0.88 | 0.92 | 0.98 |
| (0.01, 0.01) | 0.71 | 0.81 | 0.61 | 0.91 | 0.98 | 0.76 | 0.87 | 0.92 |

Table 10: Laftr-EqOdd results when decreasing minority representation for Adult dataset, sensitive attribute:$age$, selected best result per attribute

| (u-elg, u-inelg) | acc | p-acc | up-acc | DP | EqOpp0 | EqOpp1 | EqOdd | sens-acc |
|---|---|---|---|---|---|---|---|---|
| (0.5, 0.5) | 0.71 | 0.74 | 0.69 | 0.96 | 0.99 | 0.92 | 0.96 | 0.52 |
| (0.1, 0.1) | 0.69 | 0.71 | 0.66 | 0.95 | 0.98 | 0.9 | 0.94 | 0.52 |
| (0.01, 0.01) | 0.67 | 0.77 | 0.56 | 0.86 | 0.96 | 0.72 | 0.84 | 0.52 |

Table 11: Laftr-EqOpp1 results when decreasing minority representation for Adult dataset, sensitive attribute:$age$, selected best result per attribute

| (u-elg, u-inelg) | acc | p-acc | up-acc | DP | EqOpp0 | EqOpp1 | EqOdd | sens-acc |
|---|---|---|---|---|---|---|---|---|
| (0.5, 0.5) | 0.72 | 0.75 | 0.7 | 0.96 | 0.99 | 0.91 | 0.95 | 0.53 |
| (0.1, 0.1) | 0.69 | 0.72 | 0.66 | 0.95 | 0.98 | 0.9 | 0.94 | 0.52 |
| (0.01, 0.01) | 0.67 | 0.77 | 0.56 | 0.86 | 0.97 | 0.72 | 0.84 | 0.52 |

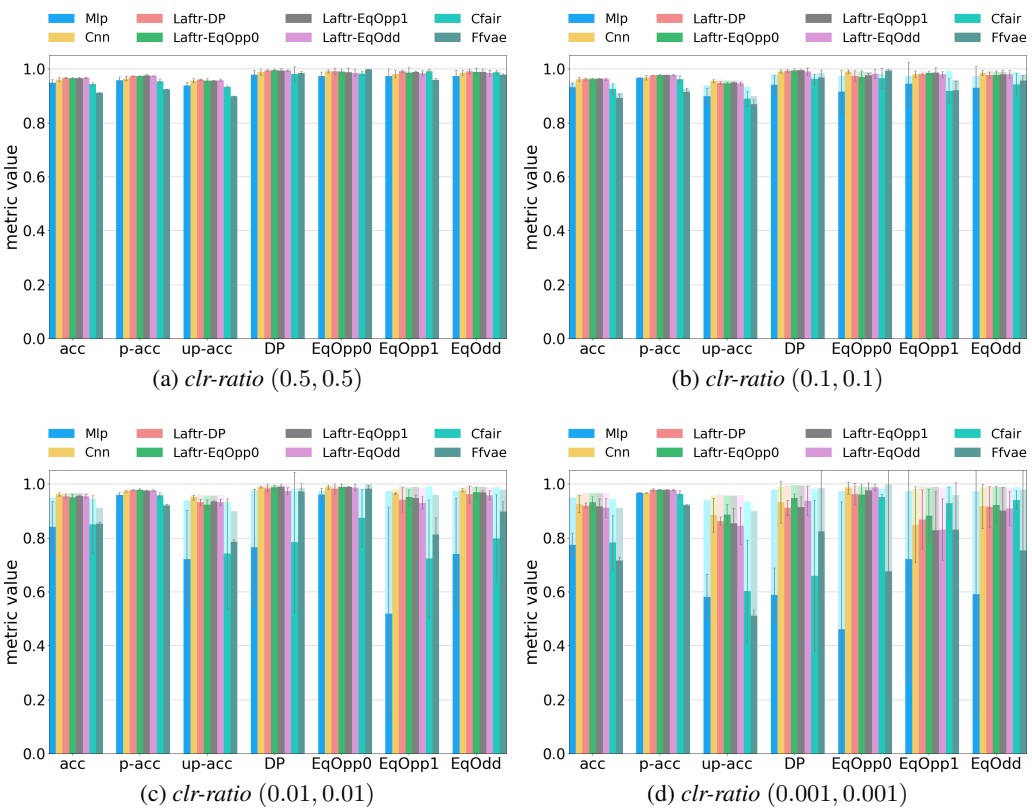

Figure 6: Comparing different models while decreasing minority representation for CI-MNIST dataset. In sub-figures 6(b), 6(c), and 6(d) the pale colors show the decrease in performance compared to the balanced case in 6(a).

Table 12: Laftr-EqOpp0 results when decreasing minority representation for Adult dataset, sensitive attribute:$age$, selected best result per attribute

| (u-elg, u-inelg) | acc | p-acc | up-acc | DP | EqOpp0 | EqOpp1 | EqOdd | sens-acc |
|---|---|---|---|---|---|---|---|---|
| (0.5, 0.5) | 0.72 | 0.75 | 0.69 | 0.96 | 1.0 | 0.91 | 0.96 | 0.52 |
| (0.1, 0.1) | 0.69 | 0.71 | 0.66 | 0.95 | 0.98 | 0.91 | 0.95 | 0.52 |
| (0.01, 0.01) | 0.66 | 0.76 | 0.56 | 0.85 | 0.97 | 0.71 | 0.84 | 0.52 |

Table 13: Laftr-DP results when decreasing minority representation for Adult dataset, sensitive attribute:$age$, selected best result per attribute

| (u-elg, u-inelg) | acc | p-acc | up-acc | DP | EqOpp0 | EqOpp1 | EqOdd | sens-acc |
|---|---|---|---|---|---|---|---|---|
| (0.5, 0.5) | 0.71 | 0.74 | 0.69 | 0.97 | 0.99 | 0.92 | 0.96 | 0.60 |
| (0.1, 0.1) | 0.69 | 0.71 | 0.66 | 0.96 | 0.99 | 0.91 | 0.95 | 0.52 |
| (0.01, 0.01) | 0.66 | 0.76 | 0.55 | 0.87 | 0.97 | 0.72 | 0.84 | 0.55 |

Table 14: Mlp results when decreasing minority representation for CI-MNIST dataset, sensitive attribute:$bck$, selected best result per attribute

| clr-ratio | acc | p-acc | up-acc | DP | EqOpp0 | EqOpp1 | EqOdd | sens-acc |
|---|---|---|---|---|---|---|---|---|
| (0.5, 0.5) | 0.95 | 0.96 | 0.94 | 0.98 | 0.97 | 0.97 | 0.97 | 0.91 |
| (0.1, 0.1) | 0.94 | 0.97 | 0.9 | 0.94 | 0.91 | 0.94 | 0.93 | 0.99 |
| (0.01, 0.01) | 0.84 | 0.96 | 0.72 | 0.76 | 0.96 | 0.52 | 0.74 | 0.9 |
| (0.001, 0.001) | 0.77 | 0.97 | 0.58 | 0.59 | 0.46 | 0.72 | 0.59 | 0.67 |

Table 15: Cnn results when decreasing minority representation for CI-MNIST dataset, sensitive attribute:$bck$, selected best result per attribute

| clr-ratio | acc | p-acc | up-acc | DP | EqOpp0 | EqOpp1 | EqOdd | sens-acc |
|---|---|---|---|---|---|---|---|---|
| (0.5, 0.5) | 0.96 | 0.96 | 0.96 | 0.99 | 0.99 | 0.98 | 0.98 | 0.56 |
| (0.1, 0.1) | 0.96 | 0.97 | 0.95 | 0.99 | 0.99 | 0.98 | 0.98 | 0.5 |
| (0.01, 0.01) | 0.96 | 0.97 | 0.95 | 0.99 | 0.99 | 0.96 | 0.97 | 0.5 |
| (0.001, 0.001) | 0.93 | 0.97 | 0.88 | 0.93 | 0.98 | 0.85 | 0.92 | 0.5 |

Table 16: Cfair results when decreasing minority representation for CI-MNIST dataset, sensitive attribute:$bck$, selected best result per attribute

| clr-ratio | acc | p-acc | up-acc | DP | EqOpp0 | EqOpp1 | EqOdd | sens-acc |
|---|---|---|---|---|---|---|---|---|
| (0.5, 0.5) | 0.94 | 0.95 | 0.93 | 0.98 | 0.98 | 0.99 | 0.98 | 1.0 |
| (0.1, 0.1) | 0.93 | 0.96 | 0.89 | 0.96 | 0.97 | 0.92 | 0.95 | 1.0 |
| (0.01, 0.01) | 0.85 | 0.96 | 0.74 | 0.78 | 0.87 | 0.72 | 0.79 | 1.0 |
| (0.001, 0.001) | 0.78 | 0.96 | 0.6 | 0.66 | 0.95 | 0.93 | 0.94 | 1.0 |

Table 17: Ffvae results when decreasing minority representation for CI-MNIST dataset, sensitive attribute:$bck$, selected best result per attribute

| clr-ratio | acc | p-acc | up-acc | DP | EqOpp0 | EqOpp1 | EqOdd | sens-acc |
|---|---|---|---|---|---|---|---|---|
| (0.5, 0.5) | 0.91 | 0.92 | 0.9 | 0.98 | 1.0 | 0.96 | 0.98 | 1.0 |
| (0.1, 0.1) | 0.89 | 0.91 | 0.87 | 0.97 | 0.99 | 0.92 | 0.96 | 1.0 |
| (0.01, 0.01) | 0.85 | 0.92 | 0.79 | 0.97 | 0.98 | 0.81 | 0.9 | 1.0 |
| (0.001, 0.001) | 0.72 | 0.92 | 0.51 | 0.82 | 0.68 | 0.83 | 0.76 | 1.0 |

Table 18: Laftr-EqOdd results when decreasing minority representation for CI-MNIST dataset, sensitive attribute:$bck$, selected best result per attribute

| clr-ratio | acc | p-acc | up-acc | DP | EqOpp0 | EqOpp1 | EqOdd | sens-acc |
|---|---|---|---|---|---|---|---|---|
| (0.5, 0.5) | 0.96 | 0.97 | 0.96 | 0.99 | 0.99 | 0.98 | 0.98 | 1.0 |
| (0.1, 0.1) | 0.96 | 0.98 | 0.95 | 0.99 | 0.98 | 0.98 | 0.98 | 0.96 |
| (0.01, 0.01) | 0.96 | 0.98 | 0.93 | 0.97 | 0.99 | 0.93 | 0.96 | 0.8 |
| (0.001, 0.001) | 0.91 | 0.98 | 0.84 | 0.94 | 0.99 | 0.83 | 0.91 | 0.8 |

Table 19: Laftr-EqOpp1 results when decreasing minority representation for CI-MNIST dataset, sensitive attribute:$bck$, selected best result per attribute

| clr-ratio | acc | p-acc | up-acc | DP | EqOpp0 | EqOpp1 | EqOdd | sens-acc |
|---|---|---|---|---|---|---|---|---|
| (0.5, 0.5) | 0.97 | 0.98 | 0.96 | 0.99 | 0.99 | 0.99 | 0.99 | 0.99 |
| (0.1, 0.1) | 0.96 | 0.98 | 0.95 | 0.99 | 0.98 | 0.98 | 0.98 | 0.95 |
| (0.01, 0.01) | 0.95 | 0.97 | 0.93 | 0.99 | 0.99 | 0.95 | 0.97 | 0.85 |
| (0.001, 0.001) | 0.92 | 0.98 | 0.85 | 0.91 | 0.98 | 0.83 | 0.91 | 0.73 |

Table 20: Laftr-EqOpp0 results when decreasing minority representation for CI-MNIST dataset, sensitive attribute:$bck$, selected best result per attribute

| clr-ratio | acc | p-acc | up-acc | DP | EqOpp0 | EqOpp1 | EqOdd | sens-acc |
|---|---|---|---|---|---|---|---|---|
| (0.5, 0.5) | 0.96 | 0.97 | 0.96 | 0.99 | 0.99 | 0.99 | 0.99 | 0.99 |
| (0.1, 0.1) | 0.96 | 0.98 | 0.95 | 0.99 | 0.97 | 0.99 | 0.98 | 0.98 |
| (0.01, 0.01) | 0.95 | 0.98 | 0.92 | 0.99 | 0.99 | 0.95 | 0.97 | 0.78 |
| (0.001, 0.001) | 0.94 | 0.98 | 0.89 | 0.95 | 0.96 | 0.88 | 0.92 | 0.67 |

Table 21: Laftr-DP results when decreasing minority representation for CI-MNIST dataset, sensitive attribute:*bck*, selected best result per attribute

| clr-ratio | acc | p-acc | up-acc | DP | EqOpp0 | EqOpp1 | EqOdd | sens-acc |
|---|---|---|---|---|---|---|---|---|
| (0.5, 0.5) | 0.96 | 0.97 | 0.96 | 0.99 | 0.99 | 0.99 | 0.99 | 1.0 |
| (0.1, 0.1) | 0.96 | 0.98 | 0.95 | 0.99 | 0.97 | 0.98 | 0.97 | 1.0 |
| (0.01, 0.01) | 0.96 | 0.98 | 0.93 | 0.98 | 0.98 | 0.94 | 0.96 | 0.86 |
| (0.001, 0.001) | 0.92 | 0.98 | 0.86 | 0.91 | 0.96 | 0.87 | 0.92 | 0.69 |

## E.2 Impact of correlation of sensitive attribute with eligibility

We report the complete set of results for debiasing models of Mlp, Cfair, Ffvae, Laftr-EqOdd, Laftr-EqOpp1, Laftr-EqOpp0, and Laftr-DP, in Tables 22 to 37, corresponding to Setting 2 in Section 4 of the main paper. Each pair in *clr-ratio* column indicate $(b_e, b_o)$, which is the ratio of images with blue background for (even=qualified, odd=unqualified) data. Figures 7, 8 compare all models side-by-side.

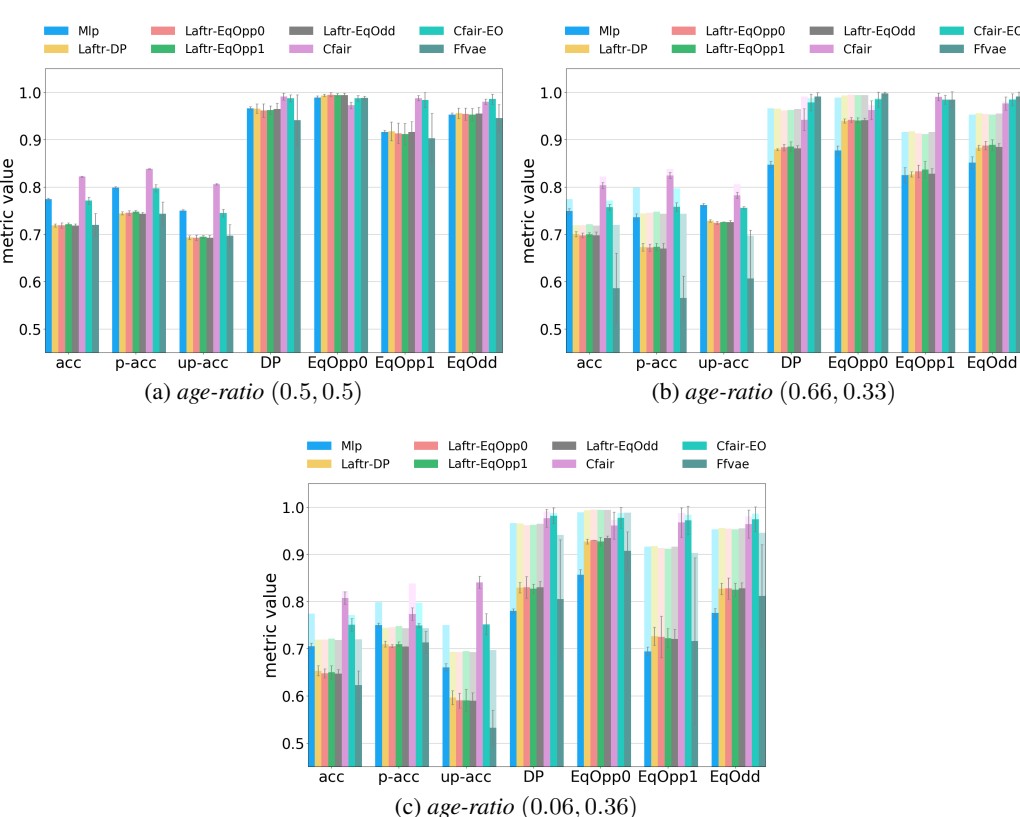

Figure 7: Comparing different models while shifting correlation of sensitive attribute ($age$) with the eligibility for Adult dataset. In sub-figures 7(b) and 7(c) the pale colors show the decrease in performance compared to the balanced case in 7(a).

Table 22: Mlp results on correlation of sensitive attribute ($age$) and eligibility for Adult dataset, selected best result per attribute

| (u-elg, u-inelg) | acc | p-acc | up-acc | DP | EqOpp0 | EqOpp1 | EqOdd | sens-acc |
|---|---|---|---|---|---|---|---|---|
| (0.5, 0.5) | 0.78 | 0.8 | 0.75 | 0.97 | 0.99 | 0.92 | 0.96 | 0.66 |
| (0.66, 0.33) | 0.75 | 0.74 | 0.76 | 0.85 | 0.88 | 0.83 | 0.85 | 0.65 |
| (0.06, 0.36) | 0.71 | 0.75 | 0.66 | 0.78 | 0.86 | 0.69 | 0.77 | 0.65 |

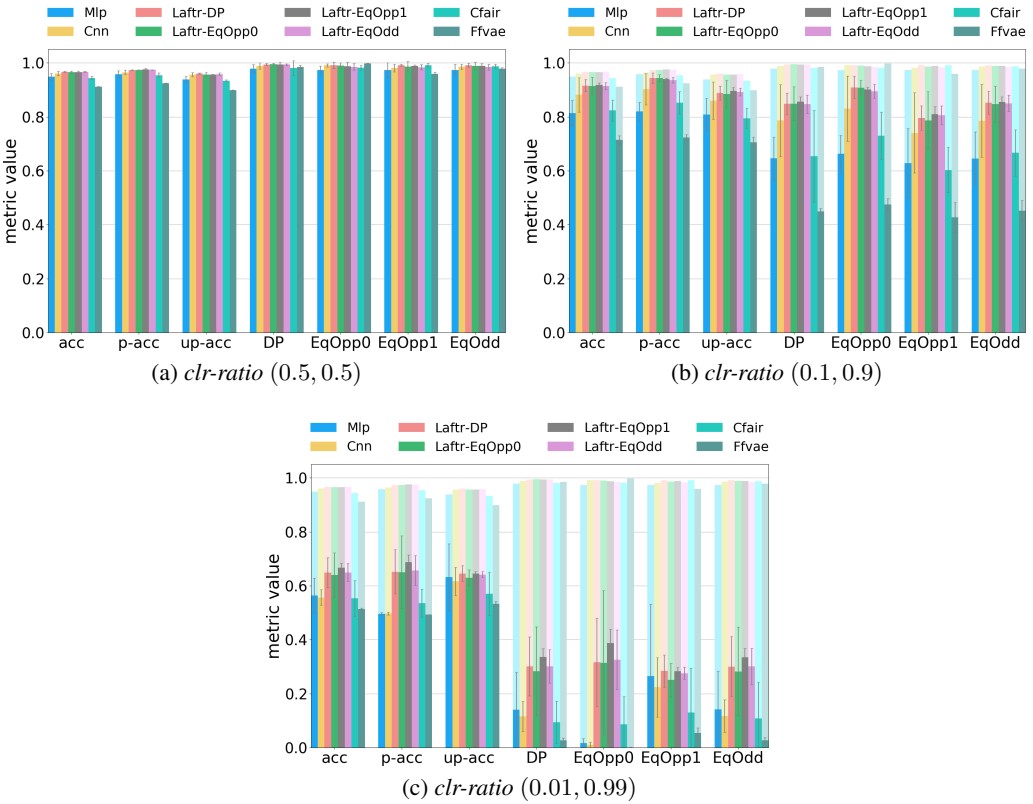

Figure 8: Comparing different models while shifting correlation of sensitive attribute ($bck$) and the eligibility for CI-MNIST dataset. In sub-figures 8(b) and 8(c) the pale colors show the decrease in performance compared to the balanced case in 8(a).

Table 23: Cfair results on correlation of sensitive attribute ($age$) and eligibility for Adult dataset, selected best result per attribute

| (u-elg, u-inelg) | acc | p-acc | up-acc | DP | EqOpp0 | EqOpp1 | EqOdd | sens-acc |
|---|---|---|---|---|---|---|---|---|
| (0.5, 0.5) | 0.82 | 0.84 | 0.81 | 0.99 | 0.97 | 0.99 | 0.98 | 0.54 |
| (0.66, 0.33) | 0.8 | 0.82 | 0.78 | 0.94 | 0.96 | 0.99 | 0.97 | 0.64 |
| (0.06, 0.36) | 0.8 | 0.77 | 0.84 | 0.98 | 0.96 | 0.97 | 0.96 | 0.54 |

Table 24: Cfair-EO results on correlation of sensitive attribute ($age$) and eligibility for Adult dataset, selected best result per attribute

| (u-elg, u-inelg) | acc | p-acc | up-acc | DP | EqOpp0 | EqOpp1 | EqOdd | sens-acc |
|---|---|---|---|---|---|---|---|---|
| (0.5, 0.5) | 0.78 | 0.8 | 0.75 | 0.99 | 0.99 | 0.98 | 0.98 | 0.52 |
| (0.66, 0.33) | 0.76 | 0.76 | 0.76 | 0.98 | 0.99 | 0.98 | 0.98 | 0.54 |
| (0.06, 0.36) | 0.75 | 0.75 | 0.75 | 0.98 | 0.98 | 0.97 | 0.97 | 0.53 |

Table 25: Ffvae results on correlation of sensitive attribute ($age$) and eligibility for Adult dataset, selected best result per attribute

| (u-elg, u-inelg) | acc | p-acc | up-acc | DP | EqOpp0 | EqOpp1 | EqOdd | sens-acc |
|---|---|---|---|---|---|---|---|---|
| (0.5, 0.5) | 0.72 | 0.74 | 0.7 | 0.94 | 0.99 | 0.9 | 0.95 | 0.93 |
| (0.66, 0.33) | 0.59 | 0.57 | 0.61 | 0.99 | 1.0 | 0.98 | 0.99 | 0.92 |
| (0.06, 0.36) | 0.62 | 0.71 | 0.53 | 0.81 | 0.91 | 0.72 | 0.81 | 0.98 |

Table 26: Laftr-EqOdd results on correlation of sensitive attribute ($age$) and eligibility for Adult dataset, selected best result per attribute

| (u-elg, u-inelg) | acc | p-acc | up-acc | DP | EqOpp0 | EqOpp1 | EqOdd | sens-acc |
|---|---|---|---|---|---|---|---|---|
| (0.5, 0.5) | 0.71 | 0.74 | 0.69 | 0.96 | 0.99 | 0.92 | 0.96 | 0.52 |
| (0.66, 0.33) | 0.7 | 0.67 | 0.73 | 0.88 | 0.94 | 0.83 | 0.89 | 0.54 |
| (0.06, 0.36) | 0.65 | 0.7 | 0.59 | 0.83 | 0.93 | 0.72 | 0.82 | 0.47 |

Table 27: Laftr-EqOpp1 results on correlation of sensitive attribute ($age$) and eligibility for Adult dataset, selected best result per attribute

| (u-elg, u-inelg) | acc | p-acc | up-acc | DP | EqOpp0 | EqOpp1 | EqOdd | sens-acc |
|---|---|---|---|---|---|---|---|---|
| (0.5, 0.5) | 0.72 | 0.75 | 0.7 | 0.96 | 0.99 | 0.91 | 0.95 | 0.53 |
| (0.66, 0.33) | 0.7 | 0.67 | 0.73 | 0.89 | 0.94 | 0.84 | 0.89 | 0.55 |
| (0.06, 0.36) | 0.65 | 0.71 | 0.59 | 0.83 | 0.93 | 0.72 | 0.82 | 0.47 |

Table 28: Laftr-EqOpp0 results on correlation of sensitive attribute ($age$) and eligibility for Adult dataset, selected best result per attribute

| (u-elg, u-inelg) | acc | p-acc | up-acc | DP | EqOpp0 | EqOpp1 | EqOdd | sens-acc |
|---|---|---|---|---|---|---|---|---|
| (0.5, 0.5) | 0.72 | 0.75 | 0.69 | 0.96 | 1.0 | 0.91 | 0.96 | 0.52 |
| (0.66, 0.33) | 0.7 | 0.67 | 0.72 | 0.88 | 0.94 | 0.83 | 0.89 | 0.55 |
| (0.06, 0.36) | 0.65 | 0.71 | 0.59 | 0.83 | 0.93 | 0.73 | 0.83 | 0.48 |

Table 29: Laftr-DP results on correlation of sensitive attribute ($age$) and eligibility for Adult dataset, selected best result per attribute

| (u-elg, u-inelg) | acc | p-acc | up-acc | DP | EqOpp0 | EqOpp1 | EqOdd | sens-acc |
|---|---|---|---|---|---|---|---|---|
| (0.5, 0.5) | 0.71 | 0.74 | 0.69 | 0.97 | 0.99 | 0.92 | 0.96 | 0.6 |
| (0.66, 0.33) | 0.7 | 0.67 | 0.73 | 0.88 | 0.94 | 0.83 | 0.89 | 0.57 |
| (0.06, 0.36) | 0.66 | 0.71 | 0.6 | 0.83 | 0.93 | 0.73 | 0.83 | 0.55 |

Table 30: Mlp results on correlation of sensitive attribute ($bck$) and eligibility for CI-MNIST dataset, selected best result per attribute

| clr-ratio | acc | p-acc | up-acc | DP | EqOpp0 | EqOpp1 | EqOdd | sens-acc |
|---|---|---|---|---|---|---|---|---|
| (0.5, 0.5) | 0.95 | 0.96 | 0.94 | 0.98 | 0.97 | 0.97 | 0.97 | 0.91 |
| (0.1, 0.9) | 0.81 | 0.82 | 0.81 | 0.65 | 0.66 | 0.63 | 0.65 | 0.96 |
| (0.01, 0.99) | 0.56 | 0.5 | 0.63 | 0.14 | 0.02 | 0.27 | 0.15 | 0.98 |

Table 31: Cnn results on correlation of sensitive attribute ($bck$) and eligibility for CI-MNIST dataset, selected best result per attribute

| clr-ratio | acc | p-acc | up-acc | DP | EqOpp0 | EqOpp1 | EqOdd | sens-acc |
|---|---|---|---|---|---|---|---|---|
| (0.5, 0.5) | 0.96 | 0.96 | 0.96 | 0.99 | 0.99 | 0.98 | 0.98 | 0.56 |
| (0.1, 0.9) | 0.88 | 0.9 | 0.86 | 0.79 | 0.83 | 0.74 | 0.78 | 0.85 |
| (0.01, 0.99) | 0.56 | 0.5 | 0.62 | 0.12 | 0.01 | 0.22 | 0.12 | 1.0 |

Table 32: Cfair results on correlation of sensitive attribute ($bck$) and eligibility for CI-MNIST dataset, selected best result per attribute

| clr-ratio | acc | p-acc | up-acc | DP | EqOpp0 | EqOpp1 | EqOdd | sens-acc |
|---|---|---|---|---|---|---|---|---|
| (0.5, 0.5) | 0.94 | 0.95 | 0.93 | 0.98 | 0.98 | 0.99 | 0.98 | 1.0 |
| (0.1, 0.9) | 0.82 | 0.85 | 0.79 | 0.65 | 0.73 | 0.6 | 0.67 | 1.0 |
| (0.01, 0.99) | 0.55 | 0.54 | 0.57 | 0.09 | 0.09 | 0.13 | 0.11 | 1.0 |

Table 33: Ffvae results on correlation of sensitive attribute (*bck*) and eligibility for CI-MNIST dataset, selected best result per attribute

| clr-ratio | acc | p-acc | up-acc | DP | EqOpp0 | EqOpp1 | EqOdd | sens-acc |
|---|---|---|---|---|---|---|---|---|
| (0.5, 0.5) | 0.91 | 0.92 | 0.9 | 0.98 | 1.0 | 0.96 | 0.98 | 1.0 |
| (0.1, 0.9) | 0.71 | 0.72 | 0.71 | 0.45 | 0.48 | 0.43 | 0.45 | 1.0 |
| (0.01, 0.99) | 0.51 | 0.49 | 0.53 | 0.03 | 0.0 | 0.05 | 0.03 | 1.0 |

Table 34: Laftr-EqOdd results on correlation of sensitive attribute (*bck*) and eligibility for CI-MNIST dataset, selected best result per attribute

| clr-ratio | acc | p-acc | up-acc | DP | EqOpp0 | EqOpp1 | EqOdd | sens-acc |
|---|---|---|---|---|---|---|---|---|
| (0.5, 0.5) | 0.96 | 0.97 | 0.96 | 0.99 | 0.99 | 0.98 | 0.98 | 1.0 |
| (0.1, 0.9) | 0.92 | 0.94 | 0.89 | 0.85 | 0.89 | 0.81 | 0.85 | 0.98 |
| (0.01, 0.99) | 0.65 | 0.66 | 0.64 | 0.3 | 0.33 | 0.28 | 0.31 | 0.97 |

Table 35: Laftr-EqOpp1 results on correlation of sensitive attribute (*bck*) and eligibility for CI-MNIST dataset, selected best result per attribute

| clr-ratio | acc | p-acc | up-acc | DP | EqOpp0 | EqOpp1 | EqOdd | sens-acc |
|---|---|---|---|---|---|---|---|---|
| (0.5, 0.5) | 0.97 | 0.98 | 0.96 | 0.99 | 0.99 | 0.99 | 0.99 | 0.99 |
| (0.1, 0.9) | 0.92 | 0.94 | 0.9 | 0.86 | 0.9 | 0.81 | 0.85 | 0.97 |
| (0.01, 0.99) | 0.67 | 0.69 | 0.65 | 0.34 | 0.39 | 0.28 | 0.34 | 0.97 |

Table 36: Laftr-EqOpp0 results on correlation of sensitive attribute (*bck*) and eligibility for CI-MNIST dataset, selected best result per attribute

| clr-ratio | acc | p-acc | up-acc | DP | EqOpp0 | EqOpp1 | EqOdd | sens-acc |
|---|---|---|---|---|---|---|---|---|
| (0.5, 0.5) | 0.96 | 0.97 | 0.96 | 0.99 | 0.99 | 0.99 | 0.99 | 0.99 |
| (0.1, 0.9) | 0.91 | 0.94 | 0.88 | 0.85 | 0.91 | 0.79 | 0.85 | 0.94 |
| (0.01, 0.99) | 0.64 | 0.65 | 0.63 | 0.28 | 0.31 | 0.25 | 0.28 | 0.96 |

Table 37: Laftr-DP results on correlation of sensitive attribute (*bck*) and eligibility for CI-MNIST dataset, selected best result per attribute

| clr-ratio | acc | p-acc | up-acc | DP | EqOpp0 | EqOpp1 | EqOdd | sens-acc |
|---|---|---|---|---|---|---|---|---|
| (0.5, 0.5) | 0.96 | 0.97 | 0.96 | 0.99 | 0.99 | 0.99 | 0.99 | 1.0 |
| (0.1, 0.9) | 0.92 | 0.94 | 0.89 | 0.85 | 0.91 | 0.8 | 0.85 | 1.0 |
| (0.01, 0.99) | 0.65 | 0.65 | 0.65 | 0.3 | 0.32 | 0.28 | 0.3 | 0.99 |

## E.3 Impact of correlation of non-sensitive attribute with eligibility

We report the complete set of results for debiasing models of Mlp, Cnn, Cfair, Ffvae, Laftr-EqOdd, Laftr-EqOpp1, Laftr-EqOpp0, and Laftr-DP, in Tables 38 to 45, corresponding to the experimental setup described in Setting 3 of Secton 4 in the main paper. Each pair in *pos-ratio* column indicate $(l_e, l_o)$, which specifies the ratio of images with box on left side for (even=eligible, odd=ineligible). Figure 9 compare all models side-by-side.

Table 38: Mlp results on correlation of non-sensitive attribute (*pos*) and eligibility for CI-MNIST dataset, selected best result per attribute

| pos | acc | p-acc | up-acc | DP | EqOpp0 | EqOpp1 | EqOdd | sens-acc |
|---|---|---|---|---|---|---|---|---|
| (0.5, 0.5) | 0.95 | 0.96 | 0.94 | 0.98 | 0.97 | 0.97 | 0.97 | 0.91 |
| (0.75, 0.25) | 0.94 | 0.95 | 0.93 | 0.97 | 0.98 | 0.94 | 0.96 | 0.98 |
| (0.9, 0.1) | 0.86 | 0.9 | 0.83 | 0.96 | 0.9 | 0.96 | 0.93 | 1.0 |

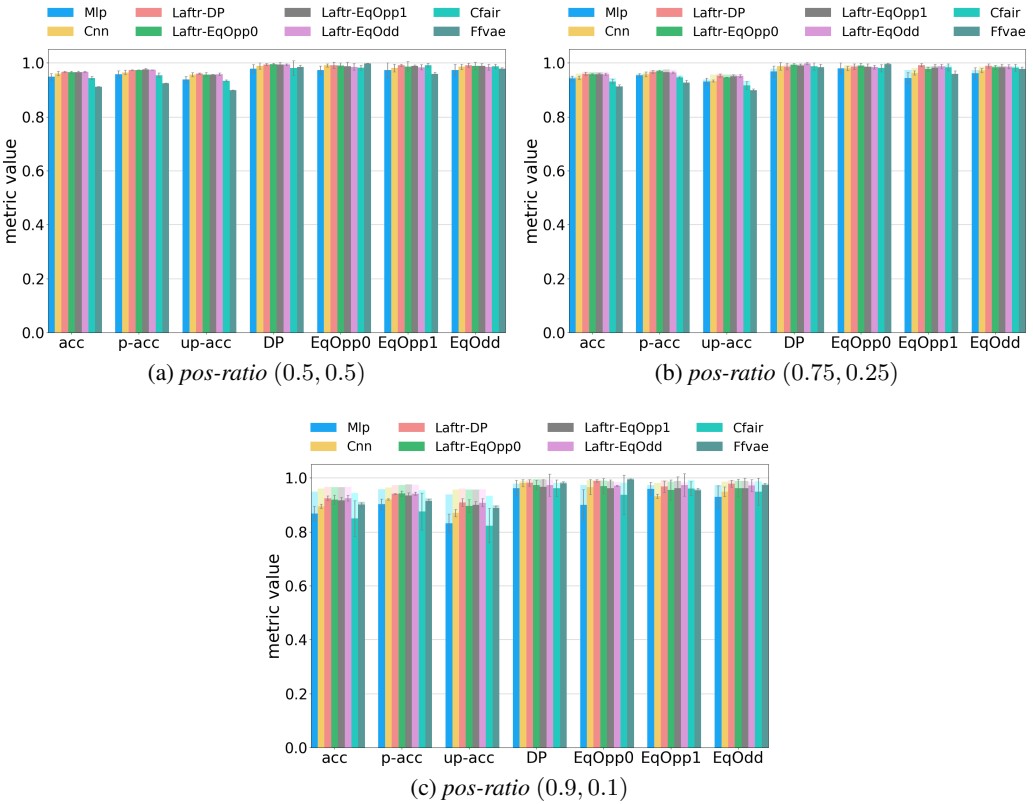

Figure 9: Comparing different models while shifting correlation of a non-sensitive attribute and the eligibility for CI-MNIST dataset. In sub-figures 9(b) and 9(c) the pale colors show the decrease in performance compared to the balanced case in 9(a).

Table 39: Cnn results on correlation of non-sensitive attribute (*pos*) and eligibility for CI-MNIST dataset, selected best result per attribute

| pos | acc | p-acc | up-acc | DP | EqOpp0 | EqOpp1 | EqOdd | sens-acc |
|------|------|-------|--------|------|--------|--------|-------|----------|
| (0.5, 0.5) | 0.96 | 0.96 | 0.96 | 0.99 | 0.99 | 0.98 | 0.98 | 0.56 |
| (0.75, 0.25) | 0.95 | 0.96 | 0.93 | 0.99 | 0.98 | 0.96 | 0.97 | 0.57 |
| (0.9, 0.1) | 0.9 | 0.92 | 0.87 | 0.98 | 0.97 | 0.93 | 0.95 | 0.63 |

Table 40: Cfair results on correlation of non-sensitive attribute (*pos*) and eligibility for CI-MNIST dataset, selected best result per attribute

| pos | acc | p-acc | up-acc | DP | EqOpp0 | EqOpp1 | EqOdd | sens-acc |
|------|------|-------|--------|------|--------|--------|-------|----------|
| (0.5, 0.5) | 0.94 | 0.95 | 0.93 | 0.98 | 0.98 | 0.99 | 0.98 | 1.0 |
| (0.75, 0.25) | 0.94 | 0.95 | 0.92 | 0.99 | 0.98 | 0.98 | 0.98 | 1.0 |
| (0.9, 0.1) | 0.85 | 0.88 | 0.82 | 0.96 | 0.94 | 0.96 | 0.95 | 1.0 |

Table 41: Ffvae results on correlation of non-sensitive attribute (*pos*) and eligibility for CI-MNIST dataset, selected best result per attribute

| pos | acc | p-acc | up-acc | DP | EqOpp0 | EqOpp1 | EqOdd | sens-acc |
|------|------|-------|--------|------|--------|--------|-------|----------|
| (0.5, 0.5) | 0.91 | 0.92 | 0.9 | 0.98 | 1.0 | 0.96 | 0.98 | 1.0 |
| (0.75, 0.25) | 0.92 | 0.93 | 0.9 | 0.98 | 1.0 | 0.96 | 0.98 | 1.0 |
| (0.9, 0.1) | 0.9 | 0.91 | 0.89 | 0.98 | 0.99 | 0.95 | 0.97 | 1.0 |

Table 42: Laftr-EqOdd results on correlation of non-sensitive attribute (*pos*) and eligibility for CI-MNIST dataset, selected best result per attribute

| pos | acc | p-acc | up-acc | DP | EqOpp0 | EqOpp1 | EqOdd | sens-acc |
|---|---|---|---|---|---|---|---|---|
| (0.5, 0.5) | 0.96 | 0.97 | 0.96 | 0.99 | 0.99 | 0.98 | 0.98 | 1.0 |
| (0.75, 0.25) | 0.95 | 0.96 | 0.95 | 1.0 | 0.98 | 0.99 | 0.98 | 1.0 |
| (0.9, 0.1) | 0.93 | 0.94 | 0.91 | 0.97 | 0.97 | 0.97 | 0.97 | 1.0 |

Table 43: Laftr-EqOpp1 results on correlation of non-sensitive attribute (*pos*) and eligibility for CI-MNIST dataset, selected best result per attribute

| pos | acc | p-acc | up-acc | DP | EqOpp0 | EqOpp1 | EqOdd | sens-acc |
|---|---|---|---|---|---|---|---|---|
| (0.5, 0.5) | 0.97 | 0.98 | 0.96 | 0.99 | 0.99 | 0.99 | 0.99 | 0.99 |
| (0.75, 0.25) | 0.96 | 0.97 | 0.95 | 0.99 | 0.98 | 0.98 | 0.98 | 0.99 |
| (0.9, 0.1) | 0.92 | 0.93 | 0.9 | 0.97 | 0.96 | 0.96 | 0.96 | 0.99 |

Table 44: Laftr-EqOpp0 results on correlation of non-sensitive attribute (*pos*) and eligibility for CI-MNIST dataset, selected best result per attribute

| pos | acc | p-acc | up-acc | DP | EqOpp0 | EqOpp1 | EqOdd | sens-acc |
|---|---|---|---|---|---|---|---|---|
| (0.5, 0.5) | 0.96 | 0.97 | 0.96 | 0.99 | 0.99 | 0.99 | 0.99 | 0.99 |
| (0.75, 0.25) | 0.96 | 0.97 | 0.95 | 0.99 | 0.99 | 0.98 | 0.98 | 0.99 |
| (0.9, 0.1) | 0.92 | 0.94 | 0.9 | 0.97 | 0.97 | 0.95 | 0.96 | 0.99 |

Table 45: Laftr-DP results on correlation of non-sensitive attribute (*pos*) and eligibility for CI-MNIST dataset, selected best result per attribute

| pos | acc | p-acc | up-acc | DP | EqOpp0 | EqOpp1 | EqOdd | sens-acc |
|---|---|---|---|---|---|---|---|---|
| (0.5, 0.5) | 0.96 | 0.97 | 0.96 | 0.99 | 0.99 | 0.99 | 0.99 | 1.0 |
| (0.75, 0.25) | 0.96 | 0.97 | 0.95 | 0.99 | 0.99 | 0.99 | 0.99 | 1.0 |
| (0.9, 0.1) | 0.93 | 0.94 | 0.91 | 0.98 | 0.99 | 0.97 | 0.98 | 1.0 |

## E.4  Impact of position and small features in the input images

Comparing baseline model with debiasing models of Mlp, Cfair, Ffvae, Laftr-EqOdd, Laftr-EqOpp1, Laftr-EqOpp0, and Laftr-DP, when position and a small feature of the image correlates with eligibility. Results are depicted in Figure in Tables 46 to 53, corresponding to the experimental setup described in Setting 4 of Section 4 in the main paper. Each pair in *pos-ratio* column indicate $(l_e, l_o)$, which specifies the ratio of images with box on left side for (even=eligible, odd=ineligible). Figure 10 compare all models side-by-side.

Table 46: Mlp results on correlation of sensitive attribute (*pos*) and eligibility for CI-MNIST dataset, selected best result per attribute

| pos-ratio | acc | p-acc | up-acc | DP | EqOpp0 | EqOpp1 | EqOdd | sens-acc |
|---|---|---|---|---|---|---|---|---|
| (0.5, 0.5) | 0.95 | 0.95 | 0.95 | 1.0 | 1.0 | 0.99 | 0.99 | 0.51 |
| (0.75, 0.25) | 0.94 | 0.95 | 0.93 | 0.93 | 0.95 | 0.92 | 0.94 | 0.59 |
| (0.9, 0.1) | 0.87 | 0.89 | 0.85 | 0.76 | 0.8 | 0.72 | 0.76 | 0.67 |

Table 47: Cnn results on correlation of sensitive attribute (*pos*) and eligibility for CI-MNIST dataset, selected best result per attribute

| pos-ratio | acc | p-acc | up-acc | DP | EqOpp0 | EqOpp1 | EqOdd | sens-acc |
|---|---|---|---|---|---|---|---|---|
| (0.5, 0.5) | 0.96 | 0.96 | 0.96 | 1.0 | 1.0 | 0.99 | 0.99 | 0.56 |
| (0.75, 0.25) | 0.94 | 0.95 | 0.94 | 0.94 | 0.94 | 0.94 | 0.94 | 0.79 |
| (0.9, 0.1) | 0.9 | 0.91 | 0.88 | 0.82 | 0.86 | 0.79 | 0.82 | 0.88 |

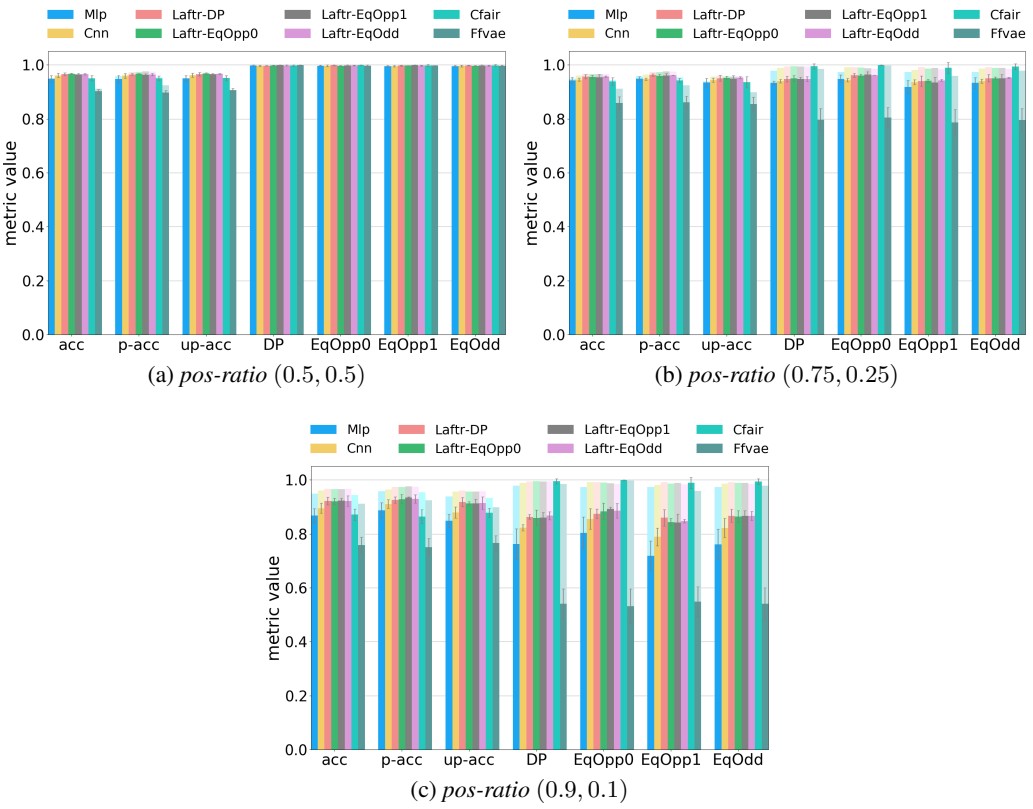

Figure 10: Impact of position and small visual components on different models' performance for CI-MNIST dataset. In sub-figures 10(b) and 10(c) the pale colors show the decrease in performance compared to the balanced case in 10(a).

Table 48: Cfair results on correlation of sensitive attribute ($pos$) and eligibility for CI-MNIST dataset, selected best result per attribute

| pos-ratio | acc | p-acc | up-acc | DP | EqOpp0 | EqOpp1 | EqOdd | sens-acc |
|---|---|---|---|---|---|---|---|---|
| (0.5, 0.5) | 0.95 | 0.95 | 0.95 | 1.0 | 1.0 | 1.0 | 1.0 | 0.82 |
| (0.75, 0.25) | 0.94 | 0.94 | 0.94 | 0.99 | 1.0 | 0.99 | 0.99 | 0.88 |
| (0.9, 0.1) | 0.87 | 0.86 | 0.88 | 0.99 | 1.0 | 0.99 | 0.99 | 0.92 |

Table 49: Ffvae results on correlation of sensitive attribute ($pos$) and eligibility for CI-MNIST dataset, selected best result per attribute

| pos-ratio | acc | p-acc | up-acc | DP | EqOpp0 | EqOpp1 | EqOdd | sens-acc |
|---|---|---|---|---|---|---|---|---|
| (0.5, 0.5) | 0.91 | 0.9 | 0.91 | 1.0 | 1.0 | 1.0 | 1.0 | 1.0 |
| (0.75, 0.25) | 0.86 | 0.86 | 0.86 | 0.8 | 0.81 | 0.79 | 0.8 | 1.0 |
| (0.9, 0.1) | 0.76 | 0.75 | 0.77 | 0.54 | 0.53 | 0.55 | 0.54 | 1.0 |

Table 50: Laftr-EqOdd results on correlation of sensitive attribute ($pos$) and eligibility for CI-MNIST dataset, selected best result per attribute

| pos-ratio | acc | p-acc | up-acc | DP | EqOpp0 | EqOpp1 | EqOdd | sens-acc |
|---|---|---|---|---|---|---|---|---|
| (0.5, 0.5) | 0.96 | 0.96 | 0.97 | 1.0 | 1.0 | 1.0 | 1.0 | 0.99 |
| (0.75, 0.25) | 0.95 | 0.96 | 0.95 | 0.95 | 0.96 | 0.94 | 0.95 | 0.64 |
| (0.9, 0.1) | 0.92 | 0.93 | 0.91 | 0.87 | 0.89 | 0.85 | 0.87 | 0.72 |

Table 51: Laftr-EqOpp1 results on correlation of sensitive attribute ($pos$) and eligibility for CI-MNIST dataset, selected best result per attribute

| pos-ratio | acc | p-acc | up-acc | DP | EqOpp0 | EqOpp1 | EqOdd | sens-acc |
|---|---|---|---|---|---|---|---|---|
| (0.5, 0.5) | 0.96 | 0.96 | 0.96 | 1.0 | 1.0 | 1.0 | 1.0 | 0.93 |
| (0.75, 0.25) | 0.95 | 0.96 | 0.95 | 0.95 | 0.96 | 0.93 | 0.95 | 0.65 |
| (0.9, 0.1) | 0.92 | 0.93 | 0.91 | 0.86 | 0.89 | 0.84 | 0.86 | 0.75 |

Table 52: Laftr-EqOpp0 results on correlation of sensitive attribute ($pos$) and eligibility for CI-MNIST dataset, selected best result per attribute

| pos | acc | p-acc | up-acc | DP | EqOpp0 | EqOpp1 | EqOdd | sens-acc |
|---|---|---|---|---|---|---|---|---|
| (0.5, 0.5) | 0.97 | 0.97 | 0.97 | 1.0 | 1.0 | 1.0 | 1.0 | 0.74 |
| (0.75, 0.25) | 0.95 | 0.96 | 0.95 | 0.95 | 0.96 | 0.94 | 0.95 | 0.61 |
| (0.9, 0.1) | 0.92 | 0.93 | 0.91 | 0.86 | 0.88 | 0.84 | 0.86 | 0.74 |

Table 53: Laftr-DP results on correlation of sensitive attribute ($pos$) and eligibility for CI-MNIST dataset, selected best result per attribute

| pos-ratio | acc | p-acc | up-acc | DP | EqOpp0 | EqOpp1 | EqOdd | sens-acc |
|---|---|---|---|---|---|---|---|---|
| (0.5, 0.5) | 0.96 | 0.96 | 0.97 | 1.0 | 1.0 | 1.0 | 1.0 | 1.0 |
| (0.75, 0.25) | 0.95 | 0.96 | 0.95 | 0.95 | 0.96 | 0.94 | 0.95 | 0.95 |
| (0.9, 0.1) | 0.93 | 0.93 | 0.92 | 0.86 | 0.87 | 0.86 | 0.86 | 0.97 |

### E.5 Impact of seed

In Figures 11 and 12 we illustrate the standard deviation of all models for all of the experiments of Adult and CI-MNIST datasets described in Section 4 of the main paper.

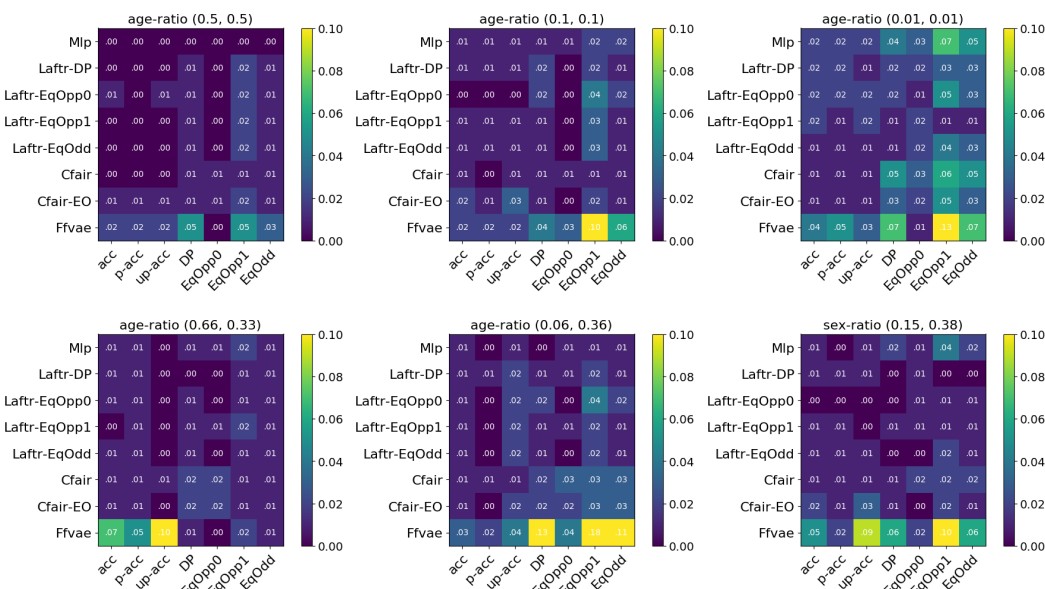

Figure 11: Standard deviation of different fairness metrics ($x$-axis) in different models ($y$-axis) over three seeds for Adult dataset. Each plot corresponds to a different experimental setup presented in Section 4.

### E.6 Correlation between dataset features and model's prediction.

In Figure 13 we present Spearman Correlation plots for each dataset and each setting of the experiments presented in Section 4. Please check Section 5.1 of the main paper for the corresponding section.

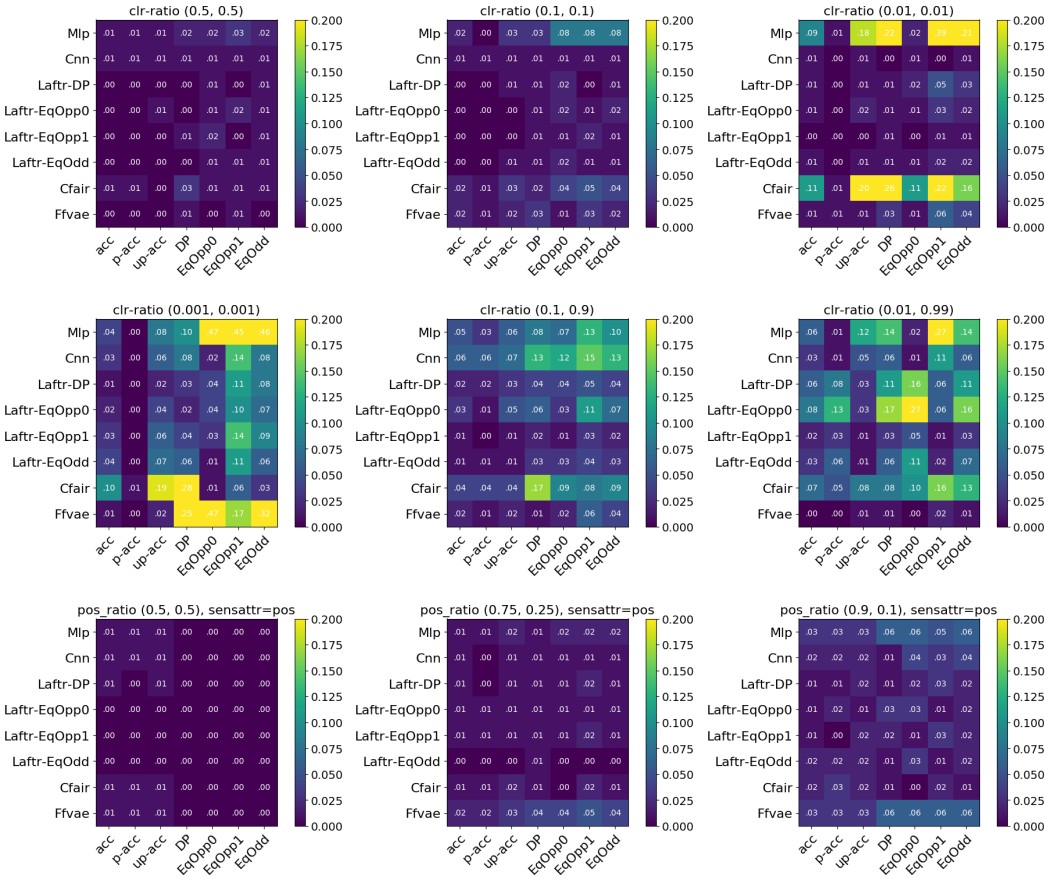

Figure 12: Standard deviation of different fairness metrics ($x$-axis) in different models ($y$-axis) over three seeds for CI-MNIST dataset. Each plot corresponds to a different experimental setup presented in Section 4.

## E.7 Impact of small population bias

Table 54 presents results for Cnn and Table 55 for Laftr-EqOpp0, where clr-ratios is kept at (0.001, 0.001) but the total dataset size has changed from $x$ to $10x$, $100x$ and $1000x$. Refer to Small percentage of the unprivileged group part in Section 5.2 of the main text.

Table 54: Cnn results for measuring whether the bias is due to small ratio or small number of samples.

| clr-ratio | acc | p-acc | up-acc | DP | EqOpp0 | EqOpp1 | EqOdd | sens-acc |
|---|---|---|---|---|---|---|---|---|
| (0.001, 0.001) x | 0.93 | 0.97 | 0.88 | 0.93 | 0.98 | 0.85 | 0.92 | 0.5 |
| (0.001, 0.001) 10x | 0.96 | 0.98 | 0.95 | 0.99 | 0.98 | 0.96 | 0.97 | 0.51 |
| (0.001, 0.001) 100x | 0.9 | 0.98 | 0.82 | 0.95 | 0.9 | 0.79 | 0.84 | 0.52 |
| (0.001, 0.001) 1000x | 0.9 | 0.98 | 0.82 | 0.95 | 0.89 | 0.78 | 0.83 | 0.52 |

Table 55: Laftr-EqOpp0 results for measuring whether the bias is due to small ratio or small number of samples.

| clr-ratio | acc | p-acc | up-acc | DP | EqOpp0 | EqOpp1 | EqOdd | sens-acc |
|---|---|---|---|---|---|---|---|---|
| (0.001, 0.001) x | 0.94 | 0.98 | 0.89 | 0.95 | 0.96 | 0.88 | 0.92 | 0.67 |
| (0.001, 0.001) 10x | 0.91 | 0.98 | 0.83 | 0.94 | 0.81 | 0.9 | 0.85 | 0.5 |
| (0.001, 0.001) 100x | 0.94 | 0.98 | 0.89 | 0.99 | 0.91 | 0.9 | 0.91 | 0.54 |
| (0.001, 0.001) 1000x | 0.96 | 0.99 | 0.93 | 0.99 | 0.95 | 0.93 | 0.94 | 0.54 |

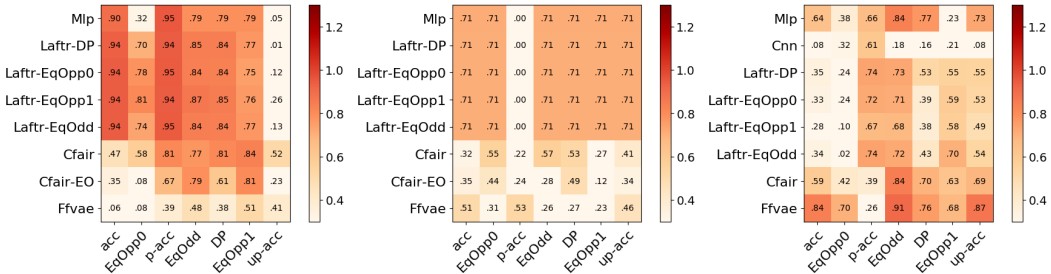

(a) Adult, Setting 1, showing correlation for *age-ratio* attribute with fairness metrics

(b) Adult, Setting 2, showing correlation for *age-ratio* attribute with fairness metrics

(c) CI-MNIST, Setting 1, showing correlation for *clr-ratio* attribute with fairness metrics

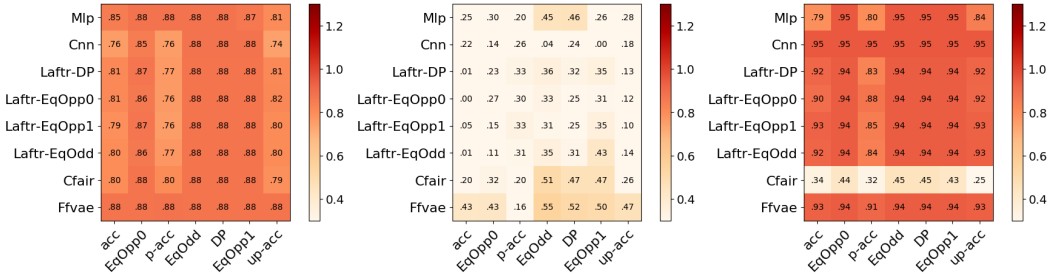

(d) CI-MNIST, Setting 2, showing correlation for *clr-ratio* attribute with fairness metrics

(e) CI-MNIST, Setting 3, showing correlation for *pos-ratio* attribute with fairness metrics

(f) CI-MNIST, Setting 4, showing correlation for *pos-ratio* attribute with fairness metrics

Figure 13: Each plot depicts correlation of one dataset attribute with fairness metrics for one setting and one dataset in Section 4. On the Adult dataset, we depict the correlation of *age-ratio* with fairness metrics as this attribute has been the sensitive feature that is changed in the experiments. On CI-MNIST , in Settings 1 and 2, we depict *clr-ratio*, and in Settings 3 and 4, we show *pos-ratio*, hence showing only the feature that is changed from the balanced case. Note that contrary to other cases, in Setting 3 *pos-ratio* is not the sensitive attribute, and background is the sensitive attribute. We plot the absolute Spearman correlation metric, where we use absolute difference of the dataset attribute from the balanced case (0.5) as input to the Spearman function. This is because numbers such as 1 and 0 have a similar meaning as they are equally away from the balanced case. Finally, we report absolute averaged correlation values over all cases. Values range in $[0, 1]$, where one indicates maximum correlation. Almost all bias-mitigation models suffer from not mitigating the strong correlation between the overall accuracy and the sensitive attribute.

## E.8 Merging bias-mitigation algorithms.

Tables 56 and 57 show results for merging Cfair and Ffvae and Tables 58 and 59 show results for merging Laftr and Ffvae.

To merge Ffvae with Laftr we added to Ffvae objective in Eq.(8), the $\mathcal{L}_{DP}^{Laftr}$ term in Eq.(3), yielding

$$\mathcal{L}_{DP}^{Ffvae-Laftr} = \mathcal{L}_{\text{Ffvae}}(p,q) - \eta \mathcal{L}_{DP}^{Laftr} \tag{10}$$

Similarly, to merge Ffvae with Cfair we added to Ffvae objective in Eq.(8), the $\mathcal{L}_{DP}^{Cfair}$ term in Eq.(7), yielding

$$\mathcal{L}_{DP}^{Ffvae-Cfair} = \mathcal{L}_{\text{Ffvae}}(p,q) - \eta \mathcal{L}_{DP}^{Cfair} \tag{11}$$

where $\eta$ is a hyper-parameter, balancing the two losses. In both cases, the added loss ($\mathcal{L}_{DP}^{Laftr}$ or $\mathcal{L}_{DP}^{Cfair}$) is applied to non-sensitive latent $z$ of Ffvae model. Please check Section 5.1 of the main paper for the discussion on the obtained results.

Table 56: Merged Ffvae and Cfair results when decreasing minority representation for Adult dataset, sensitive attribute:$age$. Added $\mathcal{L}_{DP}^{Cfair}$ to Eq.(8). Compare with Ffvae Table 9 and Cfair Table 7 results.

| (u-elg, u-inelg) | acc | p-acc | up-acc | DP | EqOpp0 | EqOpp1 | EqOdd | sens-acc |
|---|---|---|---|---|---|---|---|---|
| (0.5, 0.5) | 0.77 | 0.81 | 0.73 | 0.99 | 0.99 | 0.97 | 0.98 | 0.92 |
| (0.1, 0.1) | 0.75 | 0.78 | 0.72 | 0.99 | 1.0 | 0.99 | 0.99 | 0.98 |
| (0.01, 0.01) | 0.72 | 0.83 | 0.62 | 0.94 | 1.0 | 0.85 | 0.93 | 0.79 |

Table 57: Merged Ffvae and Cfair results on correlation of sensitive attribute ($age$) and eligibility for Adult dataset. Added $\mathcal{L}_{DP}^{Cfair}$ to Eq.(8). Compare with Ffvae Table 25 and Cfair Table 23 results.

| (u-elg, u-inelg) | acc | p-acc | up-acc | DP | EqOpp0 | EqOpp1 | EqOdd | sens-acc |
|---|---|---|---|---|---|---|---|---|
| (0.5, 0.5) | 0.77 | 0.81 | 0.73 | 0.99 | 0.99 | 0.97 | 0.98 | 0.92 |
| (0.66, 0.33) | 0.74 | 0.73 | 0.75 | 1.0 | 1.0 | 1.0 | 1.0 | 0.92 |
| (0.06, 0.36) | 0.7 | 0.76 | 0.64 | 0.98 | 0.99 | 0.96 | 0.97 | 0.97 |

Table 58: Merged Ffvae and Laftr-DP results when decreasing minority representation for Adult dataset, sensitive attribute:$age$. Added $\mathcal{L}_{DP}^{Laftr}$ to Eq.(8). Compare with Ffvae Table 9 and Laftr-DP Table 13 results.

| (u-elg, u-inelg) | acc | p-acc | up-acc | DP | EqOpp0 | EqOpp1 | EqOdd | sens-acc |
|---|---|---|---|---|---|---|---|---|
| (0.5, 0.5) | 0.66 | 0.71 | 0.61 | 0.99 | 1.0 | 0.98 | 0.99 | 0.93 |
| (0.1, 0.1) | 0.6 | 0.67 | 0.54 | 1.0 | 1.0 | 1.0 | 1.0 | 0.98 |
| (0.01, 0.01) | 0.6 | 0.69 | 0.52 | 1.0 | 1.0 | 1.0 | 1.0 | 0.92 |

Table 59: Merged Ffvae and Laftr-DP results on correlation of sensitive attribute ($age$) and eligibility for Adult dataset. Added $\mathcal{L}_{DP}^{Laftr}$ to Eq.(8). Compare with Ffvae Table 25 and Laftr-DP Table 29 results.

| (u-elg, u-inelg) | acc | p-acc | up-acc | DP | EqOpp0 | EqOpp1 | EqOdd | sens-acc |
|---|---|---|---|---|---|---|---|---|
| (0.5, 0.5) | 0.66 | 0.71 | 0.61 | 0.99 | 1.0 | 0.98 | 0.99 | 0.93 |
| (0.66, 0.33) | 0.49 | 0.5 | 0.49 | 1.0 | 1.0 | 1.0 | 1.0 | 0.93 |
| (0.06, 0.36) | 0.6 | 0.69 | 0.51 | 0.97 | 0.99 | 0.95 | 0.97 | 0.98 |

