# OpenReview forum: "Benchmarking Bias Mitigation Algorithms in Representation Learning through Fairness Metrics"
_NeurIPS.cc/2021/Track/Datasets_and_Benchmarks/Round1 — NeurIPS 2021 Datasets and Benchmarks Track (Round 1)_

### Official Review · Reviewer_b3N6 · 2021-07-04
**Good paper but I wish the results were presented better**

**Rating:** 7
**Confidence:** 4
**Correctness:** Yes, they appear correct to me.

**Strengths:**

Overall, I like the paper. It does a good job of designing different datasets and stress testing common fairness methods. The recognize and appreciate the amount of work that goes into working on a paper like this. I also really enjoyed reading the discussion section - it’s a vital component of this paper and lays out opportunities for future research. I think having such a benchmark would be beneficial for the community.


**Weaknesses:**

I like the paper but find it hard to appreciate how much the findings “actually matter”. The paper shows that bias-mitigation methods don’t work as well as dataset bias increases. But by what magnitude? How biased do datasets have to be to render bias-mitigation algorithms fail? What I think is missing is a stronger grounding between the dataset design choices and real world use cases. Let me explain this in a bit more detail:

What makes the results, and therefore the experiments section, hard to appreciate is the difficulty in deciphering to what degree existing bias-mitigation methods actually fail. What does it actually mean to test with an age-ratio of (0.06, 0.36)? Or with clr-ratio of (0.01, 0.99)? Without a clear explanation of how these ratios map onto ecologically valid situations that might arise in real world use cases, it’s hard to appreciate why it matters.

On a similar vein, the raw numbers of all these fairness metrics are fine to report. But I wish the paper instead presented a relative decrease in these metrics from the balanced training set scenario. I wish the main paper had a plot where the x axis was the ratio change and the y axis was the relative decrease in the fairness metric from a balanced training dataset. A relative change, with an explanation of what that decrease implies, would have made the results much more impactful.


**Additional Feedback:**

NA

**Clarity:**

Line 6: “into the working 6 of these methods”. Weird phrasing.

“Eligibility” and “sensitive variable” are defined later in the methods section and are general well known amongst people who work in fairness. But it’s worth defining these terms when you first use them in the introduction section.

Figures are really hard to read. The axes and labels need to be larger in font.


**Documentation:**

Yes

**Ethics:**

I don't see newer ethical concerns introduced by this paper.

**Relation To Prior Work:**

The paper does a good job of laying out the related work.

**Summary And Contributions:**

This paper evaluates the representations learned by 3000 different deep learning fairness methods on common datasets. It finds that model bias increases as datasets become more imbalanced or when dataset attributes become more correlated. It also finds that sensitive information remains in the learned representations regardless of the bias-mitigation algorithms.

The three main findings are: (1) imbalance or scarce underrepresented groups induce bias, (2) correlations between sensitive attribute and eligibility induce bias, (3) some model sare susceptible to bias from random seed initializations

---

> ### Author Response · Authors · 2021-07-15
> **Response to Reviewer b3N6 - Part 2/2**
>
> Clarity:
>
> As suggested we made the below corrections.
>
> > Line 6: “into the working 6 of these methods”. Weird phrasing.
>
> Thanks for pointing it out. We rephrased Line 6.
>
> > “Eligibility” and “sensitive variable” are defined later in the methods section and are general well known amongst people who work in fairness. But it’s worth defining these terms when you first use them in the introduction section.
>
> We added definitions for “eligibility” and “sensitive variable” in Introduction in lines 26 to 31. Thanks for pointing it out.
>
> > Figures are really hard to read. The axes and labels need to be larger in font.
>
> We increased the font of axes and labels in the plots for better reading. We have also added the same plots with a bigger size in the Appendix.

---

> ### Author Response · Authors · 2021-07-15
> **Response to Reviewer b3N6 - Part 1/2**
>
> > I like the paper but find it hard to appreciate how much the findings “actually matter”. The paper shows that bias-mitigation methods don’t work as well as dataset bias increases. But by what magnitude? How biased do datasets have to be to render bias-mitigation algorithms fail? What I think is missing is a stronger grounding between the dataset design choices and real world use cases. Let me explain this in a bit more detail:
>
> > What makes the results, and therefore the experiments section, hard to appreciate is the difficulty in deciphering to what degree existing bias-mitigation methods actually fail. What does it actually mean to test with an age-ratio of (0.06, 0.36)? Or with clr-ratio of (0.01, 0.99)? Without a clear explanation of how these ratios map onto ecologically valid situations that might arise in real world use cases, it’s hard to appreciate why it matters.
>
> Please note that numbers such as (0.06, 0.36) or (0.01, 0.99) indicate the ratio (or percentage when multiplied in 100) of the unprivileged group in (eligible, ineligible) classes. More importantly, they indicate the degree of correlation or imbalance under which the evaluation is applied. For example, (0.1, 0.1) or (0.01,0.01) indicate an imbalanced setting where the unprivileged group is a minority in both eligible and ineligible groups, or (0.1, 0.9) indicates a correlated setting where the unprivileged group is a minority in eligible group and majority in ineligible groups, in which case eligibility is correlated with the sensitive attribute.
>
> To provide examples in real-world datasets of such ratios, in CelebA [g], for example, we have (0.05, 0.80) ratio of young people in (gray hair, no gray hair) groups, (0.05, 0.8) ratio of males in (lipstick, no lipstick) groups, and (0.2, 0.95) ratio of old people in (no gray hair, gray hair) groups. These examples have similar ratios as the correlated case mentioned above. In COMPAS dataset [i], we have (0.08, 0.07) ratio of Hispanics in (male, female) groups, and (0.005, 0.001) ratio of Asians in (male, female) groups. In German credit dataset [k], we have (0.1, 0.1) ratio of people older than 52 with (good, bad) credits and (0.01, 0.01) ratios of people older than 66 with (good, bad) credits. These examples have a similar ratio compared to the imbalanced cases mentioned above. In Visual Question Answering (VQA), as shown in references [a, b, c], the datasets cover very diverse cases, ranging from cases with very high to a very low probability of observing questions in a particular category, which in turn causes bias. Such imbalanced or correlated scenarios commonly happen in real-world datasets as datasets usually have heavy-tailed distributions [d, e, f]. Hence, the evaluation of the robustness of models to such cases is not unrealistic and can happen in real-world applications.
>
> [a] Aishwarya Agrawal, Aniruddha Kembhavi, Dhruv Batra, Devi Parikh. C-VQA: A compositional split of the visual question answering (vqa) v1. 0 dataset. CVPR 2018.
>
> [b] Aishwarya Agrawal, Dhruv Batra, Devi Parikh. Analyzing the behavior of visual question answering models. EMNLP 2016.
>
> [c] Sainandan Ramakrishnan, Aishwarya Agrawal, Stefan Lee. Overcoming language priors in visual question answering with adversarial regularization. NeurIPS 2018.
>
> [d] Wanli Ouyang, Xiaogang Wang, Cong Zhang, Xiaokang Yang. Factors in finetuning deep model for object detection with long-tail distribution. CVPR 2016.
>
> [e] Jimei Yang, Brian Price, Scott Cohen, Ming-Hsuan Yang. Context driven scene parsing with attention to rare classes. CVPR 2014.
>
> [f] Dora Zhao Angelina Wang Olga Russakovsky. Understanding and Evaluating Racial Biases in Image Captioning. 2021.
>
> [g] Liu, Ziwei et al. “Deep Learning Face Attributes in the Wild.” 2015 IEEE International Conference on Computer Vision (ICCV) 2015.
>
> [i] Angwin, J., Larson, J., Mattu, S., and Kirchner, L. Machine bias: There’s software used across the country to predict future criminals. and it’s biased against blacks. ProPublica, 2016.
>
> [k] German credit data set. https://archive.ics.uci.edu/ml/support/statlog+(german+credit+data).
>
>
> > On a similar vein, the raw numbers of all these fairness metrics are fine to report. But I wish the paper instead presented a relative decrease in these metrics from the balanced training set scenario. I wish the main paper had a plot where the x axis was the ratio change and the y axis was the relative decrease in the fairness metric from a balanced training dataset. A relative change, with an explanation of what that decrease implies, would have made the results much more impactful.
>
> Thanks for the suggestion. We have added new bar plots (in pale colors), which show the relative decrease of metrics compared to the balanced training dataset. Please check the paper for new plots. For better readability and comparisons between the plots, we did not consider ratio change for the x-axis and only updated the y axis on the degree of performance change.

---

### Official Review · Reviewer_jHz5 · 2021-07-06
**Useful synthetic dataset & benchmarking procedure for evaluating fairness-enforcing dl methods.(modified: 19 Jul 2021)**

**Rating:** 6
**Confidence:** 2

**Strengths:**

Relevance: The synthetic dataset is a simple and effective proxy for simulating the class imbalance and correlation to sensitive variables in imaging. It is very understandable and easily communicated.

The standardization of evaluation is a valuable contribution to allow researchers to compare models fairly, something that seems to be lacking in papers to date. It is especially valuable then that the code is freely available and appears to be well documented and detailed, and that there is some attention paid to the affect of random seeds (especially given the results reported in previous benchmarking studies, such as Locatello, Bauer et al 2019).

A thorough study of the SOTA models under progressively more challenging conditions, illustrating where there is a failure to ensure both fairness and higher accuracy.

There are comparisons between methods that are disentangling vs methods that impose fairness on the representation. Authors make a novel contribution by combining those models to achieve an incremental improvement in the sota for fairness metrics.

A discussion section contains interesting insights into the standard fairness models. For example, it shows that bias can be caused by data scarcity. As another example, it proposes a novel method for determining the extent of sensitive information representation in the latent space.


**Weaknesses:**

Significance: having a real world data set is a strength of the paper. But a limitation is that it is not an imaging set. So there may be  doubts about whether a model tested under this framework is shown to be 'fair' in a more general, real-world imaging data setting (though of course it is useful for non-imaging settings).

Significance: a current limitation of the synthetic dataset is that (at least by default), there are only 2 groups - 'privileged' and 'underprivileged' group. Using only 2 groups is reasonable for a first study, but future work may want to go into more groups, and it would be useful if CI-MNIST can easily handle this case.

Relevance: the chosen models are mostly adversarial methods (and a vae), but other methods are referenced in the paper that have a reasonable number of citations (this may actually be fine if an argument is made that these methods are the most important methods).




**Additional Feedback:**

Figure 1 has a column called "after shade", but that isn't explained in the figure caption, and the word "shade" does not appear in the main document.

**Clarity:**

The paper is written logically, and is clear for the most part.

There are some errors of expression, for example 86, 169-170, 391-393.

Paragraph starting 143 is a bit hard to follow. It's a little wordy, and uses the concepts of 'sensitive', 'priveleged', and 'eligibility' in quick succession in a way that was a bit confusing. It could be phrased more simply.

**Correctness:**

The synthetic dataset is constructed in a simple and logical way.
The evaluation methods are reasonable in that they simply use those metrics that are used in the current state-of-the-art models.
My checking of the codebase suggests the implementation is good, but this was a limited exploration.

**Documentation:**

The github repo is documented well enough to reproduce the datasets and re-run the experiments done for the paper submission.

**Ethics:**

No concerns to raise.

**Relation To Prior Work:**

The categorisation of models into pre-processing, in-processing, post-processing is clear.

It's made clear that this work is focusing on DL approaches (in particular adversarial methods), so as to be distinct from previous reviews on non-DL methods.

This reviewer does not have deep-enough knowledge of this field to judge whether the models that are chosen for the empirical study really are the SOTA. There is also not enough knowledge to say whether the related work section is missing important papers.

**Summary And Contributions:**

1) A synthetic image dataset, CI-MNIST (Correlated and Imbalanced MNIST), where each data point has a group membership and a sensitive variable.  Users can set parameters to (i) control relative group size to represent **class imbalance**; and (ii) create **correlations** between a variable (that is sensitive or non-sensitive) and an outcome/eligibility variable.
Similarly, they propose a modification to an existing real dataset (Adult) so that users can experiment with imbalance and correlation to sensitive variables, though this real data set is not imaging.

2) A protocol / procedure that (i) generates the above data sets with varying degrees of class imbalance and correlations (between sensitive & outcome variables), and (ii) evaluates the SOTA bias-correction models under standard fairness evaluation metrics, for the datasets generated as described. They do detailed discussion of results and comparisons between current best models.

3) Well-documented code for (i) generating the datasets that are in contribution, (ii) implementing the SOTA models in fairness, and (iii) evaluating models under standard evaluation metrics.

---

> ### Author Response · Authors · 2021-07-15
> **Response to Reviewer jHz5**
>
> > Significance: having a real world data set is a strength of the paper. But a limitation is that it is not an imaging set. So there may be doubts about whether a model tested under this framework is shown to be 'fair' in a more general, real-world imaging data setting (though of course it is useful for non-imaging settings).
>
> We chose the Adult dataset, since it was one of the key datasets (if not the most used dataset) in the fairness community [38, 6, 7, 9, 23, 24, 25, 27, 28, 33] and we wanted to analyze to what extent the results obtained on our synthetic dataset (in which we can systematically control features and evaluate multiple settings easily) carries over to this widely used real dataset. Although there are some imaging-based datasets [3, 8, 25] used in the fairness community, we are not sure if this modality is the most important one as other modalities [2, 3, 4, 5] have also been used. We chose Adults as it was more widely used in the community. More importantly, our results in Section 4 (more specifically in settings 1 and 2) show the results observed on our synthetic dataset (image-based) and the real Adult dataset (numeric based), which have different input modalities, have similar trends. This indicates the analyzed settings not just carry over across synthetic and real datasets, but also across modalities. Using these datasets helps us observe results on different modalities and understand how much different modalities impact the results. (Please note that the given reference numbers are based on the newly submitted version of the paper and not the old one.)
>
> > Significance: a current limitation of the synthetic dataset is that (at least by default), there are only 2 groups - 'privileged' and 'underprivileged' group. Using only 2 groups is reasonable for a first study, but future work may want to go into more groups, and it would be useful if CI-MNIST can easily handle this case.
>
> Thanks for the suggestion. We agree that enabling multiple sensitive groups is useful for future work in this direction. We have added a feature in our codebase to include multiple sensitive groups to the CI-MNIST dataset (such as using multiple background colors or multiple positions of the boxes).
>
> > Relevance: the chosen models are mostly adversarial methods (and a vae), but other methods are referenced in the paper that have a reasonable number of citations (this may actually be fine if an argument is made that these methods are the most important methods).
>
> We have chosen some of the most important (well-cited) in-processing deep learning methods. Due to the adversarial nature of most bias-mitigation deep learning models [6, 7, 8, 9, 26, 27, 28, 29, 30, 31, 32, 33], we focused on this category of models. The models we evaluate are built upon some methods referenced in the Related Works section. For example, FFVAE [8], which is evaluated in our paper, is built upon FactorVAE [29], beta-TCVAE [30], Fair VAE [25], and VAE [34]. Our evaluated models are also closely related to some methods referenced in the Related Works section. Marx et al. [28] disentangles sensitive latent representation from the non-sensitive latent representation using an autoencoder, which is similar to FFVAE [8] that we evaluate in our framework. Models introduced in [27,32, 33] use an adversarial network to remove latent sensitive information similar to LAFTR [6]. We didn't use MUBAL [9] as we used this model initially but found its training very unstable, since it directly applies adversarial learning on the target class labels (and not the latent representation). (Please note that the given reference numbers are based on the newly submitted version of the paper and not the old one.)
>
> Clarity:
>
> > There are some errors of expression, for example 86, 169-170, 391-393.
>
> We have corrected the above errors in the main paper. Thanks for pointing this out.
>
> > Paragraph starting 143 is a bit hard to follow. It's a little wordy, and uses the concepts of 'sensitive', 'priveleged', and 'eligibility' in quick succession in a way that was a bit confusing. It could be phrased more simply.
>
> We have explained it in a better way in the main paper. Thanks for pointing this out.
>
> Additional Feedback:
>
> > Figure 1 has a column called "after shade", but that isn't explained in the figure caption, and the word "shade" does not appear in the main document.
>
> Thanks for pointing it out, we have modified the figure and the caption.

---

### Official Review · Reviewer_2RFS · 2021-07-06
**A solid benchmark that may contribute to the fairness community**

**Rating:** 7
**Confidence:** 4

**Strengths:**

- The authors release the code to contribute to the fairness community.
- The authors provide a comprehensive examination on 3 fair representation learning algorithms. Their observations could motivate researchers to seek for theoretical explanations.

**Weaknesses:**

Although the authors claim that their benchmark is designed for a wide variety of scenarios, it seems that their benchmark only supports binary labels and binary sensitive attributes (if I got wrong, please correct me). The fact that it doesn't support multiple protected groups or even continuous sensitive attributes narrows down the application and the contribution of this work. I am also concerning about whether their benchmark could be generalized to multi-class classification or regression.

**Additional Feedback:**

The authors have addressed all my concerns. Especially, they added a feature to enable multiple sensitive groups. It will definitely broaden the comparison of constrained optimization algorithms and enhance the significance to the fairness community, which is appreciated. Therefore, I would raise my rating to 7 (the original rating is 6).

------
Questions:

- Why does controlling the position of small boxes help to control the correlation? This needs more details.
- Do you consider controlling the imbalance of both labels $Y$ and sensitive attributes $S$?



**Clarity:**

The paper is well written and easy to follow up. The rationale for evaluation setup is clearly explained. Although, there are some typos.

**Correctness:**

The evaluation methods and experiment design are appropriate and performed correctly.

**Documentation:**

There is sufficient detail to support reproducibility. The authors release the Github repo to their benchmark.

**Ethics:**

I don't think there are any ethical concerns in this work.

**Relation To Prior Work:**

This work discussed how this work differs from previous contributions. The authors emphasis that they evaluate *deep learning-based models*.  I recognize their novelty in fairness benchmark but don't think evaluating deep learning models is rather novel.

**Summary And Contributions:**

This paper articulates one important problem in the current fairness community, that *the inconsistencies in the experimentation and dataset setups hinders a fair comparison of bias-mitigating methods*. To remedy this, this paper introduces a framework to benchmark fair representation learning algorithms in a wide variety of scenarios with controllable imbalanced labels and correlated sensitive attributes. The authors release the codebase to contribute to the fairness community, which is appreciated. They also discuss how the different factors may affect the fairness evaluation and the potential sources of bias.

---

> ### Author Response · Authors · 2021-07-15
> **Response to Reviewer 2RFS**
>
> > Although the authors claim that their benchmark is designed for a wide variety of scenarios, it seems that their benchmark only supports binary labels and binary sensitive attributes (if I got wrong, please correct me). The fact that it doesn't support multiple protected groups or even continuous sensitive attributes narrows down the application and the contribution of this work. I am also concerning about whether their benchmark could be generalized to multi-class classification or regression.
>
> The proposed benchmark is currently designed for binary labels and binary sensitive attributes. We made this decision for the following reasons: 1) the majority of the fairness research uses binary sensitive attributes [6, 7, 9, 15, 24, 27, 41, 42, 47, 49, 50, 51] and binary target labels [6, 7, 9, 15, 24, 27, 40, 41, 42, 47, 49, 50, 51], hence we chose to apply the dominant scenario. 2) Even in this setting we had to train about 3,000 different models to make a fair comparison and analyze the results, so it was comprehensive enough as an independent study. We left other settings, which also require a reasonable amount of model training and analysis, as future directions. Please note that the given reference numbers are based on the newly submitted version of the paper and not the old one.
> As requested by the reviewer, we have added a feature to include multiple sensitive groups to our CI-MNIST dataset (such as using multiple background colors or multiple positions of the boxes) in our codebase to enable future research in this direction. It can be also modified to include multiple digits (0-9) to make it a multi-class eligibility label. Thanks for pointing it out.
>
> Additional Feedback Questions:
> >
> > Why does controlling the position of small boxes help to control the correlation? This needs more details.
>
> Pos-ratio (le, lo) attribute is the ratio for (even, odd) numbers in which the small box is on the left side of the image. For the remaining images in both even and odd categories, the small box is on the right side of the image. This allows introducing a correlation between the position of the small box position (left/right) and eligibility (even/odd) by controlling this ratio. For example, when Pos-ratio is (1, 0) for even numbers the small box is always on the left side and for odd numbers the small box is always on the right side. This creates the complete correlated case, as the location of the box in the image clarifies whether an image is even (eligible case) or odd (ineligible case).  The pos-ratio (0.5, 0.5), on the other hand, is the uncorrelated setting, as for either even or odd digits the small box is on the left side of the image, only in 50% of cases. One can control the degree of correlation by transitioning between these two extreme cases. Thanks for pointing it out. We have added more clarifications in Section 3.1.
>
> > Do you consider controlling the imbalance of both labels Y and sensitive attributes S?
>
> We only control the imbalance with respect to sensitive attributes S. However, note that there is an imbalance intrinsically in the labels Y in the Adult dataset, and imbalance in labels Y of CI-MNIST dataset can be controlled pretty easily in the provided codebase.

---

### Author Response · Authors · 2021-07-15
**Thanks to all the reviewers for your valuable feedback**

We would like to thank all the reviewers for their time and feedback. Please find the answers to the raised questions/concerns below. We have also updated the paper and used violet color to highlight the sections that answer/clarify the issues mentioned by reviewers.

---

### Decision · Program_Chairs · 2021-07-26

**Decision:**

Accept

**Comment:**

This paper introduces a framework for benchmarking fair representation learning methods that systematically controls dataset imbalances and correlations. While the toy and synthetic nature of the dataset limits the conclusions that can be drawn, reviewers agree there is significant value in having a standardized and controllable dataset for benchmarking purposes.

Reviewers are in agreement that this paper is a valuable contribution to this track and addresses an important open problem within algorithmic fairness.